# Modulation of fatty acid elongation in cockroaches sustains sexually dimorphic hydrocarbons and female attractiveness

Xiao-Jin Pei[1], Yong-Liang Fan[1]*, Yu Bai[2], Tian-Tian Bai[1], Coby Schal[3], Zhan-Feng Zhang[1], Nan Chen[2], Sheng Li[2], Tong-Xian Liu[1]*

1 State Key Laboratory of Crop Stress Biology for Arid Areas and Key Laboratory of Integrated Pest Management on Crops in Northwestern Loess Plateau, Ministry of Agriculture, Northwest A&F University, Yangling, Shaanxi, China, 2 Guangdong Provincial Key Laboratory of Insect Developmental Biology and Applied Technology and Institute of Insect Science and Technology, School of Life Sciences, South China Normal University, Guangzhou, China, 3 Department of Entomology and Plant Pathology and W.M. Keck Center for Behavioral Biology, North Carolina State University, Raleigh, North Carolina, United States of America

* yfan@nwafu.edu.cn (Y-LF); txliu@nwafu.edu.cn (T-XL)

**Data Availability Statement:** PacBio data sets have been deposited in NCBI's SRA under the accession numbers SRR9143014 and SRR9143013. All BgElo sequences have been

## Abstract

Insect cuticular hydrocarbons (CHCs) serve as important intersexual signaling chemicals and generally show variation between the sexes, but little is known about the generation of sexually dimorphic hydrocarbons (SDHCs) in insects. In this study, we report the molecular mechanism and biological significance that underlie the generation of SDHC in the German cockroach *Blattella germanica*. Sexually mature females possess more C29 CHCs, especially the contact sex pheromone precursor 3,11-DimeC29. RNA interference (RNAi) screen against the fatty acid elongase family members combined with heterologous expression of the genes in yeast revealed that both *BgElo12* and *BgElo24* were involved in hydrocarbon (HC) production, but *BgElo24* is of wide catalytic activities and is able to provide substrates for *BgElo12*, and only the female-enriched *BgElo12* is responsible for sustaining female-specific HC profile. Repressing *BgElo12* masculinized the female CHC profile, decreased contact sex pheromone level, and consequently reduced the sexual attractiveness of female cockroaches. Moreover, the asymmetric expression of *BgElo12* between the sexes is modulated by sex differentiation cascade. Specifically, male-specific *BgDsx* represses the transcription of *BgElo12* in males, while *BgTra* is able to remove this effect in females. Our study reveals a novel molecular mechanism responsible for the formation of SDHCs and also provide evidences on shaping of the SDHCs by sexual selection, as females use them to generate high levels of contact sex pheromone.

## Introduction

Sexual dimorphism is prevalent in the animal kingdom. Females and males independently evolve some traits that enhance survival and reproduction under the pressure of divergent selection forces, thus leading to sexual dimorphism of traits such as body size [1]. Under sexual

submitted to NCBI's GenBank under the accession numbers MT925720 and MW380216–MW380238.

**Funding:** This work was funded by the National Natural Science Foundation of China (http://www.nsfc.gov.cn), Grant No. 31772533 to YLF; the United States National Science Foundation (https://www.nsf.gov), IOS-1557864 to CS and the Northwest A&F University Special Foundation (https://www.nwsuaf.edu.cn), 2009-01-001-TXL to TXL. The funders had no role in study design, data collection and analysis, decision to publish, or preparation of the manuscript.

**Competing interests:** The authors have declared that no competing interests exist.

**Abbreviations:** AD, adult day; CDS, complete coding sequence; CHC, cuticular hydrocarbon; CRE, *cis*-regulatory element; Desat, desaturase; DMEM, Dulbecco's Modified Eagle Medium; *Dmrt*, *doublesex/mab-3 related*; dsRNA, double-stranded RNA; *Dsx*, *doublesx*; ELO, elongase; FA, fatty acid; FAME, fatty acid methyl ester; FAR, fatty acyl-CoA reductase; FAS, fatty acid synthase; GC–MS, gas chromatography–mass spectrometry; HC, hydrocarbon; JH, juvenile hormone; LCFA, long-chain fatty acid; PCA, principal component analysis; qPCR, quantitative PCR; RACE, 5′ Rapid Amplification of cDNA Ends; RH, relative humidity; RNAi, RNA interference; RT-PCR, real-time PCR; RT-qPCR, real-time quantitative PCR; SC-uracil, *S. cerevisiae* minimal medium minus uracil; SDCHC, sexually dimorphic cuticular hydrocarbon; SDHC, sexually dimorphic hydrocarbon; VLCFA, very long–chain fatty acid; WR, wing raising.

selection, asymmetric selection on the sexes can also result in the evolution of sexually dimorphic traits; however, these traits are subject to both inter- and intrasexual selection [2,3]. Because males and females of the same species share the majority of their genomes, the genetic basis of sex-specific traits that evolve under sexual selection is poorly understood. It is widely assumed that sexually dimorphic regulation of gene expression facilitates sex-specific adaptations [4–7]. Recruitment of preexisting genes or pathways into sexually dimorphic regulatory contexts has been proposed as a remarkable mechanism enabling the divergence of gene expression between the sexes [8–10]. The sex differentiation pathway is a conserved switch or regulator governing a set of downstream genes that direct sexually dimorphic traits [11,12]. The signaling cascades that transform sex differences (sex-specific chromosomes) into alternative splicing of sex differentiation genes (e.g., *Transformer* and *Doublesex* in insects) and the patterns of splicing these genes into sex-based isoforms have been widely described in different insect orders [13–17]. However, sexually dimorphic traits are usually generated from tightly associated multiple biosynthetic steps that are executed by a series of genes, and the nodes at which sex-determining signals connect with the biosynthetic pathway are poorly understood. Moreover, how these genes are translated into sexually dimorphic traits under elaborate spatial and temporal patterns also needs to be elucidated.

The main function of insect cuticular hydrocarbons (CHCs) is to waterproof the cuticle to resist dehydration under dry conditions [18]. In many insects, CHCs have been co-opted to serve as chemical signals (pheromones) that mediate intraspecific communication [19,20]. Sexually dimorphic cuticular hydrocarbon (SDCHC) profiles are widespread in insects [21–23], but the regulatory networks that underlie the formation of SDCHCs largely remain unknown. Considerable progress has been made toward understanding the genetic basis of hydrocarbon (HC) biosynthesis and CHC variation in insects. An acetyl-CoA carboxylase catalyzes the biosynthesis of malonyl-CoA, and a cytosolic fatty acid synthase (FAS) incorporates malonyl-CoA units onto the acetyl-CoA primer to form linear long-chain fatty acids (LCFAs). A microsomal FAS catalyzes the biosynthesis of methyl-branched LCFAs using methylmalonyl-CoAs [24–30]. The LCF acyl-CoA can be selectively desaturated by a specific fatty acid desaturase (Desat), leading to unsaturated fatty acids [31,32]. The LCF acyl-CoAs are elongated to form very long–chain fatty acids (VLCFAs) with specific chain lengths by a cyclic fatty acid elongation system, including a rate-limiting elongase (ELO) and 3 other enzymes, elongase firstly incorporate a malonyl-CoA unite into the LCF acyl-CoA to generate a ketoacyl CoA, the ketoacyl CoA is then reduced by a 3-keto-acyl-CoA-reductase to hydroxyacyl-CoA, which is dehydrated by a 3-hydroxy-acyl-CoA dehydratase to an enoyl-CoA, and further reduced by a *trans*-enoyl-CoA-reductase to yield the elongated VLCF acyl-CoA [29,33]. The VLCF acyl-CoAs are finally reduced to long-chain alcohols by fatty acyl-CoA reductases (FARs) and converted to HCs by the P450 oxidative decarbonylase CYP4Gs [34–36]. It is now clear that the variation in CHCs is primarily reflected in the chain length, number, and positions of methyl groups and the degree of unsaturation, which are determined by ELOs, FASs, and Desats, respectively [37–39]. These enzymes have been studied in few insect species with regard to sexually dimorphic hydrocarbons (SDHCs).

In the fruit fly *Drosophila melanogaster*, several studies have elucidated the pheromonal sexual dimorphism of CHCs. Female flies produce C27 and C29 dienes (7,11-heptacosadiene and 7,11-nonacosadiene), both of which function as female-specific contact sex pheromone components, whereas male flies produce C23 and C25 monoenes (7-tricosene and 7-pentacosene) [40]. Ferveur and colleagues found that the targeted expression of a sex differentiation gene, *Transformer*, in male oenocytes feminized the male CHC profile and elicited homosexual courtship from other males [41]. These findings implicated sex differentiation genes in the production of SDHCs. About 10 years later, Chertemps and colleagues revealed that the

female-specific *desatF* was responsible for the generation of pheromonal dienes in female *D. melanogaster* [31], and they also found that the sexual dimorphism in HC chain length was modulated by the female-specific *eloF* [33]. The specific expression of *desatF* in *D. melanogaster* females was due to a special *cis*-regulatory element (CRE) located upstream of *desatF* in *D. melanogaster*; the CRE presented a doublesex protein (Doublesx, Dsx) binding site that could be recognized by the female-specific isoform of *Dsx* (*Dsx-F*), and the binding of Dsx-F activated the transcription of *desatF*, whereas male-specific Dsx showed no regulatory activity, resulting in sexual dimorphism of *desatF* expression [42]. However, the regulation of other genes involved in the production of SDHCs remains unknown in *D. melanogaster*, and studies in other insects are even rarer.

The German cockroach *Blattella germanica* is a significant worldwide indoor pest [43]. Considerable work has been done on biochemical aspects of CHCs in *B. germanica* [22,44–46]. CHCs in *B. germanica* function as waterproofing agents [47], and as importantly, specific HC components are also the biosynthetic precursors for the production of contact sex pheromone components. The 3,11-DimeC29 is the most abundant CHC in females. Its hydroxylation at the 2 position, catalyzed by an age- and sex-specific putative cytochrome P450 and further oxidation of the–OH, generates the main female-specific contact sex pheromone component, 3,11-DimeC29-2-one [48]. The contact sex pheromone is an efficient courtship signal. When a sexually mature male's antennae detect the sex pheromone on the female body surface, the male will raise his wings to expose a specialized tergal gland. Nutrients in the tergal secretion engage females in feeding, and the female is placed in an appropriate position for copulation [49–51]. More recently, Wexler and colleagues decoded the alternative splicing patterns of sex differentiation genes in *B. germanica*. BgTra is only functional in females and can splice *BgDsx* into 2 nonfunctional female-specific isoforms (*BgDsx^F^*), whereas males generate a complete functional male-type *BgDsx* (*BgDsx^M^*) [52]. These findings provide opportunities for exploring the mechanisms underlying the SDHCs in *B. germanica*.

In this study, we employed *B. germanica* as a model insect. We characterize the SDCHCs, describe their temporal development, identify *BgElo* genes, and provide evidence that *BgElo12* is responsible for the formation of SDCHC profiles. In addition, we found that the female-enriched HCs are important in the generation of the female-specific contact sex pheromone; RNA interference (RNAi) of *BgElo12* in females decreased courtship responses of males. Moreover, we show that *BgElo12* is under the regulation of the sex differentiation pathway; *BgDsx^M^* can specifically repress the transcription of *BgElo12*. These findings suggest that the generation of SDHCs is achieved by placing a fatty acid elongation gene (*BgElo12*) under the regulation of *BgDsx^M^* and linking the sex differentiation regulatory cascade with the HC biosynthesis pathway, resulting in the asymmetric gene expression and SDHCs in *B. germanica*.

## Results

### SDCHCs in *B. germanica*

The temporal development of SDCHCs is rarely reported. In order to understand the molecular mechanisms of SDHC generation, we first analyzed the CHC profiles during sexual maturation. The oocytes of female cockroaches mature after eclosion by taking up vitellogenin until ovulation [45], and females become sexually receptive and mate 4 to 5 days before ovulation [53]. In our study, female cockroaches oviposited late on day 7 or early day 8; therefore, adult days 1 to 6 (AD1–6) adult cockroaches were used for CHC analysis. Different CHC components were identified as previously described [54]. We found no qualitative differences between males and females, but quantitative differences in CHCs became more apparent with adult age (S1 Table). At an early adult stage (AD1 and AD2), females and males had similar

CHC profiles. However, differences were apparent at AD3 and gradually increased until AD6 (Fig 1A). Along with sexual maturation in males, the proportions of C27 CHCs and 9-; 11-; 13-; and 15-MeC29 (male-enriched peak 17) obviously increased, while other C29 CHCs, particularly 3,7-; 3,9-; and 3,11-DimeC29 (female-enriched peak 24), significantly decreased. In females, however, the CHC profiles consistently displayed high proportions of C29 CHCs and especially 3,7-; 3,9-; and 3,11-DimeC29 (Fig 1B–1E). Principal component analysis (PCA) showed that the male and female CHC profiles were more similar at AD1, but diverged at AD6, and the divergence was mainly reflected in principal component 2, which largely represents the chain length factor (Fig 1F and 1G). The sexual dimorphism of CHCs was generated at the adult stage (nymphal CHC chromatogram showed no qualitative differences between males and females; S1 Fig), and the sexes diverged with sexual maturation. Notably, the differences between male and female CHC profiles suggested that chain length is an important factor in sexual dimorphism of CHCs. The putative HC biosynthetic pathway was shown in Fig 1H. Our previous work identified a FAS gene (*BgFas1*) and a P450 oxidative decarbonylase gene (*CYP4G19*) that are involved in HC biosynthesis, but both showed no function in the formation of sexual dimorphism of CHCs [55,56]. However, the first step in elongation of LCF acyl-CoA is the key step to determine the final chain lengths of HCs. Therefore, we next identify *BgElo* genes in *B. germanica*.

## *BgElo12* and *BgElo24* are involved in HC biosynthesis

We found that the differences of CHCs between females and males are largely reflected in carbon chain length. Based on this assumption, we searched for potential *BgElo* genes in the *B. germanica* genomic data and our full-length transcriptomic data [55,57]. A total of 24 different *BgElo* candidate genes were identified, and all *BgElos* were cloned and re-sequenced. Sequence alignment revealed that different BgElo proteins showed high homology and contained the conserved HXXHH and YXYY motifs [58]. All BgElo proteins displayed an ELO domain and several transmembrane domains (S1 Appendix).

The fatty acid precursors used for HC production generally originate in the oenocytes, but, also, can be transported from the fat body to oenocytes [29]. Therefore, we first analyzed the transcript levels of all *BgElos* in the fat body and abdominal integument. Results showed that *BgElo24* and *BgElo12* were highly expressed in the integument, and *BgElo1*, *2*, *3*, *6*, *7*, *9*, *10*, *11*, *14*, *17*, *20*, and *22* were also abundant in the integument, while other *BgElos* were nearly undetectable (Fig 2A). In the fat body, *BgElo10* and *BgElo22* were highly expressed, *BgElo1*, *2*, *3*, *11*, *12*, and *24* were slightly expressed, and other *BgElos* were undetectable (Fig 2B).

Based on these results, knockdown of the genes that were expressed in the abdominal integument or fat body was performed. A first injection of double-stranded RNA (dsRNA) was performed in early fifth-instar nymphs (N5D1 or N5D2), and a boost injection was performed 1 week later, and then the treated cockroaches were collected at different adult stages and subjected to CHC analysis. Knockdown of different *BgElo* genes significantly decreased the mRNA level (S2 Fig). Gas chromatography–mass spectrometry (GC–MS) analysis of CHCs showed that C27 CHCs were affected by many genes—knockdown of *BgElo1*, *10*, *12*, *14*, *20*, and *24* caused a significant increase in C27 CHCs, and repression of *BgElo12* showed the greatest increase of the content of C27 CHCs (Fig 2C). C28 CHCs were rarely affected, with only knockdown of *BgElo24* causing a significant decrease in C28 CHCs (Fig 2D). C29 CHCs were the most abundant in *B. germanica*, and only knockdown of *BgElo12* or *BgElo24* significantly decreased their amount (Fig 2E). Knockdown of *BgElo12* or *BgElo24* also dramatically decreased the amount of C30 CHCs, while knockdown of *BgElo2* increased C30 CHCs (Fig 2F). CHCs with chain lengths greater than 30 occur in low quantities in *B. germanica*, but the

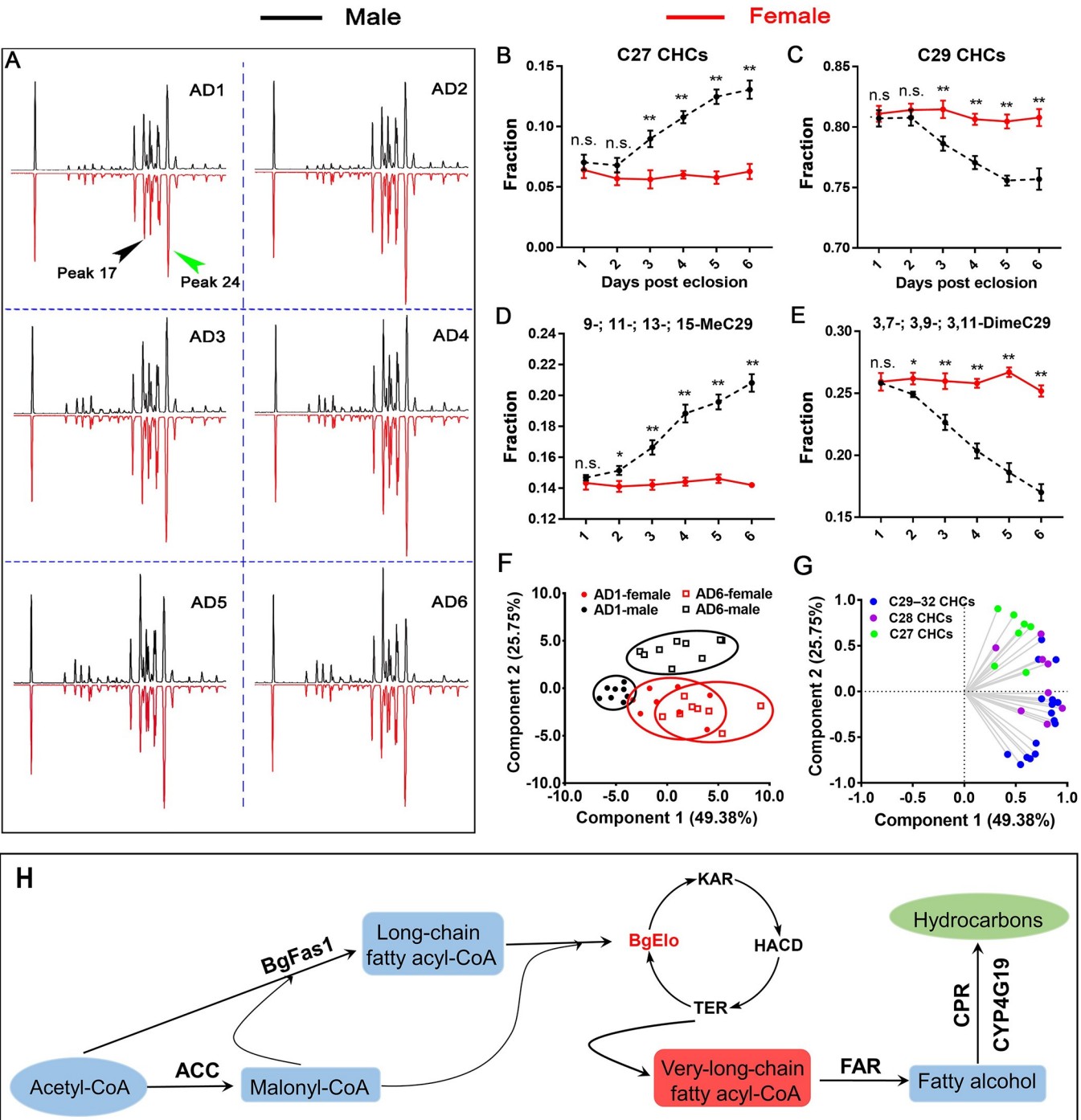

**Fig 1. Development of sexually dimorphic CHCs. (A)** Comparisons of AD1–AD6 male (black) and female (red) CHC profiles. The first peak represents the internal standard *n*-hexacosane. Other peaks correspond to the CHCs in S1 Table. Peak 17 is male enriched (9-; 11-; 13-; and 15-MeC29) and gradually increased with age in males. Peak 24 is female enriched (3,7-; 3,9-; and 3,11-DimeC29). The proportions of C27 CHCs **(B)**, C29 CHCs **(C)**, Peak 17 **(D)**, and Peak 24 **(E)** in the total CHCs from AD1 to AD6 were compared between males (black dashed lines) and females (solid red lines). Data are shown as mean ± SEM (n.s. represents no significant difference; *$P < 0.05$, **$P < 0.01$, 2-tailed Student *t* test, $n = 8$–10). **(F, G)** PCA of AD1 and AD6 CHC profiles of males and females. In **(F)**, each dot represents a datum calculated from one cockroach. AD1 male (black solid circles) and female (red solid circles) CHC profiles overlapped, whereas AD6 male (black open square) and female (red open square) CHC profiles separated along PC2. In the loading diagram **(G)**, CHCs with different chain lengths are marked by green (C27), purple (C28), and blue (C29–C32), showing that C27 CHCs vectored toward males along PC2, whereas longer chain CHCs vectored toward females. **(H)** Putative HC biosynthetic pathway in *B. germanica*. The data underlying this figure are included in S1 Table and S1 Data. ACC, acetyl-CoA carboxylase; AD, adult day; BgElo, fatty acid elongase from *B. germanica*; BgFas1, a FAS from *B. germanica* involved in

HC biosynthesis; CHC, cuticular hydrocarbon; CPR, NADPH-cytochrome P450 reductase; CYP4G19, a P450 oxidative decarbonylase from *B. germanica* involved in HC biosynthesis; FAR, fatty acyl-CoA reductase; FAS, fatty acid synthase; HACD, 3-hydroxy-acyl-CoA dehydratase; HC, hydrocarbon; KAR, 3-keto-acyl-CoA reductase; PCA, principal component analysis; TER, *trans*-enoyl-CoA reductase.

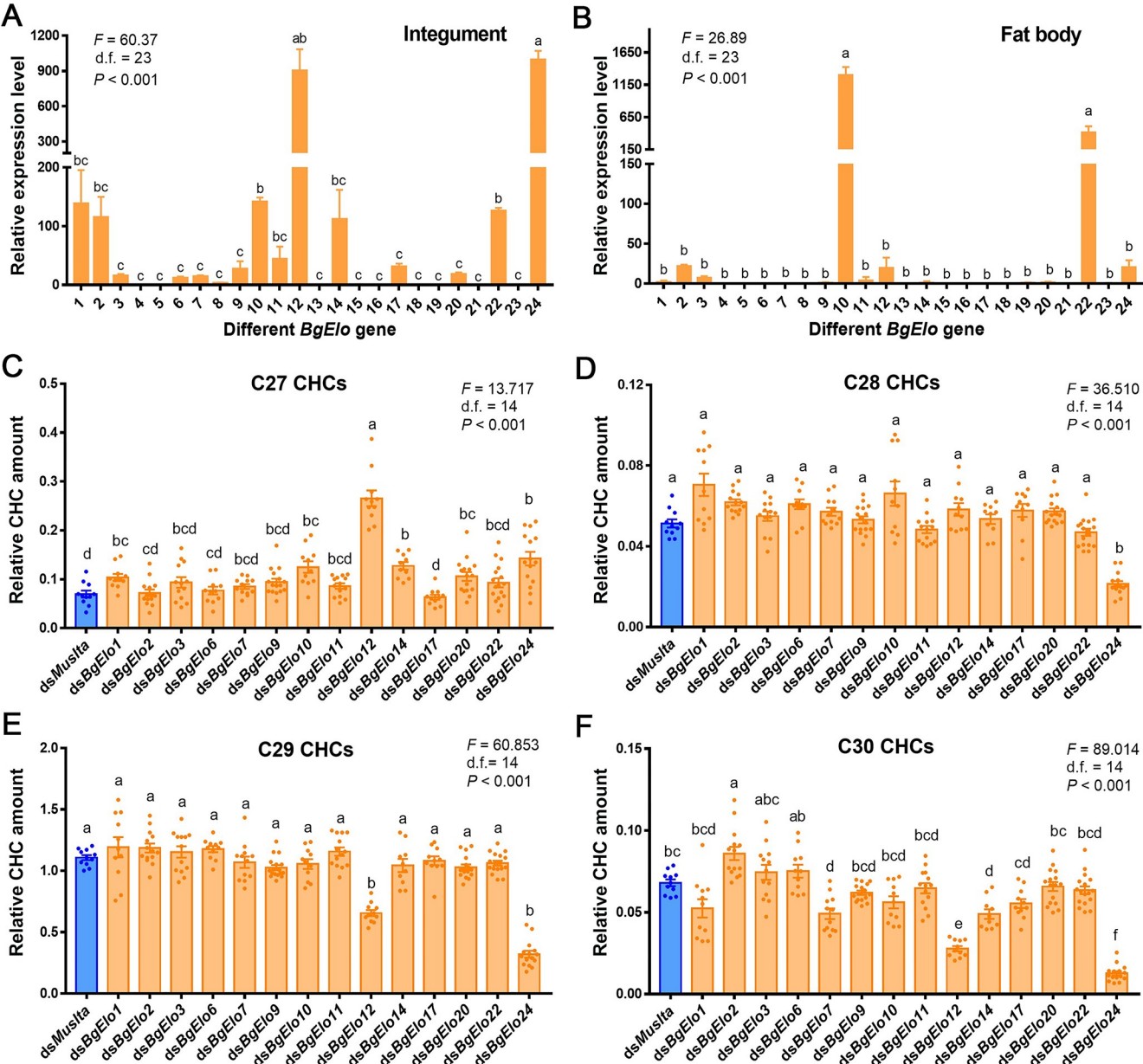

**Fig 2. RNAi screen to identify *BgElo* genes involved in long-chain CHC biosynthesis.** Transcript levels of different *BgElo* genes in the abdominal integument (**A**) and fat body (**B**). The numbers on the abscissa correspond to *BgElo1–24*. Data are shown as mean ± SEM and calculated from 4 replicates (each replicate contains 4 cockroaches for the integument and 8 cockroaches for the fat body). The influence of knockdown of different *BgElo* genes on C27 (**C**), C28 (**D**), C29 (**E**), and C30 (**F**) CHCs is shown. Data are presented as mean ± SEM and calculated from 10 to 16 AD2 female cockroaches, each dot indicating a single datum from one cockroach. Only knockdown of *BgElo12* and *BgElo24* resulted in significant increases of C27 CHCs and decreases of C29 CHCs. Different letters indicate statistically significant differences between groups using Welch ANOVA (Games–Howell multiple comparisons test, $P < 0.05$). The data underlying this figure are included in S1 Data and S2 Table. AD, adult day; CHC, cuticular hydrocarbon; RNAi, RNA interference.

most striking result was a sharp decline that resulted from *BgElo24*-RNAi (S3 Fig). The detailed changes of individual CHCs after knockdown of *BgElos* are available in S2 Table. In conclusion, these data strongly suggest that *BgElo12* and *BgElo24* are involved in CHC production, but we cannot rule out that other *BgElo* genes might have a less prominent role in C27 to C30 HC biosynthesis.

## *BgElo12* is the terminal gene in maintaining SDHC profiles

Although it appears that both *BgElo12* and *BgElo24* are involved in HC production, it is unclear which gene is responsible for sexual dimorphism of CHCs. Therefore, we compared the CHC profiles of treated female cockroaches with wild-type male cockroaches. The results showed that only RNAi of *BgElo12* made the female CHC profile more similar to the male profile (Fig 3A), reflected by a significant increase of C27 CHCs without affecting the male-enriched peak 17, and a selective down-regulation of C29 CHCs, with a dramatic down-regulation of the female-enriched peak 24 and some other C29 CHCs (Fig 3B). Also, knockdown of *BgElo12* increased the proportions of C27 CHCs and peak 17, while the proportions of C29 CHCs and peak 24 significantly decreased (Fig 3D). These changes converged the female CHC profile toward the male CHC profile. The repression of *BgElo24* down-regulated all C28 to C32 CHCs (Fig 3C), even though knockdown of *BgElo24* increased the proportion of C27 CHCs and decreased the proportion of C29 CHCs and peak 24 (Fig 3E); the amount of the male-enriched peak 17 was also dramatically down-regulated (Fig 3C). Similar results were generated in males, and RNAi of *BgElo12* generated the characteristics of male CHC profiles (S4A Fig), while *BgElo24*-RNAi down-regulated all C28 to C32 CHCs (S4B Fig). In order to confirm these results, a second RNAi target was used for both genes, and similar results were generated (S5A and S5B Fig). In addition, we analyzed the internal HCs after repression of *BgElo12* or *BgElo24*. The internal HCs underwent similar changes as CHCs (S5C and S5D Fig). These results suggest that the changes imposed by *BgElo12*- or *BgElo24*-RNAi were caused by a deficiency in de novo HC biosynthesis and not the transport of HCs from internal tissues to the cuticle.

We also examined the spatiotemporal expression of *BgElo12* and *BgElo24*. Both *BgElo12* and *BgElo 24* were primarily expressed in the abdominal integument, where the oenocytes that produce HCs are located (S6 Fig). Monitoring of *BgElo12* expression in females and males across different developmental stages showed that female and male *BgElo12* mRNA levels were similar at N6D4 (4-day-old sixth-instar nymph) and early adult stage, but its expression level was higher in females than in males at AD3, and the difference increased through AD6 (Fig 3F), a pattern similar to the production of SDCHCs. The expression levels of *BgElo24* were higher in males than in females starting at AD2 (Fig 3G). These results also support that only *BgElo12*, and not *BgElo24*, is involved in sexual dimorphism of CHCs in *B. germanica*.

Finally, because CHCs are important waterproofing agents in insects [18], we investigated the roles of *BgElo12* and *BgElo24* in desiccation resistance. We found that repression of *BgElo24* dramatically decreased tolerance of desiccation, but RNAi of *BgElo12* had little effect on desiccation tolerance (Fig 3H). These results further indicate that the biological significance of *BgElo12* in cockroaches may sustain sexual dimorphism of CHCs, whereas *BgElo24* may contribute to desiccation tolerance.

## BgElo24 provides VLCFA substrates for BgElo12

Thus far, we showed that both *BgElo12* and *BgElo24* are involved in HC production: RNAi of *BgElo12* only selectively decreased some HCs, while repression of *BgElo24* dramatically down-regulated all C28 to C32 HCs, and some HCs were affected by both *BgElo12* and *BgElo24*. We

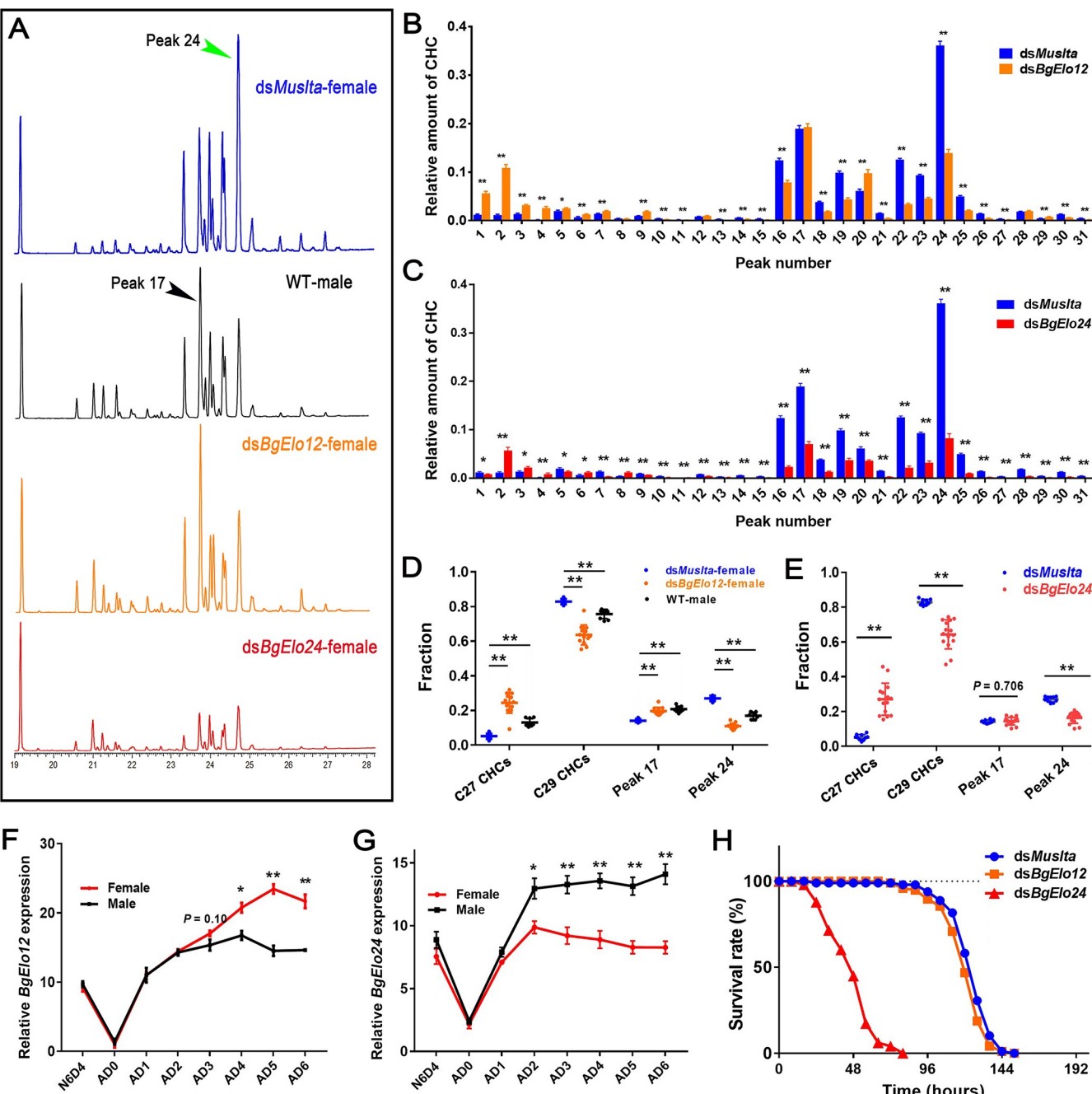

**Fig 3. *BgElo12* is involved in generating SDHCs. (A**) Gas chromatogram of CHC profiles from differently treated cockroaches. The CHC profiles of sexually mature females and males (AD6–AD8) showed a marked difference, and RNAi of only *BgElo12* in females generated a male-like CHC profile. Quantitative influence of *BgElo12*-RNAi (**B**) and *BgElo24*-RNAi (**C**) in females on the amount of each CHC (the peak numbers corresponding to S1 Data). Percentage change of the C27 CHCs, C29 CHCs, male-enriched peak 17, and female-enriched peak 24 after *BgElo12*-RNAi (**D**) or *BgElo24*-RNAi (**E**). Data in (**B–E**) are shown as mean ± SEM, *$P < 0.05$, **$P < 0.01$; differences between 2 groups were determined by 2-tailed Student *t* test, and differences among 3 groups were determined by 1-way ANOVA and LSD multiple comparisons, $n = 10$ for *dsMuslta*, 11 for *dsBgElo12*, 15 for *dsBgElo24*, and 9 for WT male. Relative expression of *BgElo12* (**F**) and *BgElo24* (**G**) among different developmental stages and different ages of adult males and females. *P* values were calculated from 4 replicates (2–3 cockroaches/replicate), **$P < 0.01$; 2-tailed Student *t* test. (**H**) Survival rates of *dsMuslta* ($n = 98$), *dsBgElo12* ($n = 96$), and *dsBgElo24* ($n = 98$) treated female cockroaches maintained at 5% RH. Survival rates were calculated every 8 hours until all cockroaches died. The data underlying this figure are included in S1 Data. CHC, cuticular hydrocarbon; LSD, least significant difference; RH, relative humidity; RNAi, RNA interference; SDHC, sexually dimorphic hydrocarbon; WT, wild-type.

also found that knockdown of *BgElo24* caused a 2-fold up-regulation of *BgElo12*, while RNAi of *BgElo12* had no significant influence on *BgElo24* mRNA level (S7A and S7B Fig). Therefore, we considered whether BgElo24 is a basic elongase that provides substrates used for HC biosynthesis and if BgElo24 generates primary precursors that can be further catalyzed by BgElo12. We verified the functions of BgElo12 and BgElo24 in VLCFA biosynthesis with heterologous expression of *BgElo12* and *BgElo24* in *Saccharomyces cerevisiae* (strain INVSc1). The genetic background of *S. cerevisiae* is relatively clear, as wild-type *S. cerevisiae* contains 3 different ELO proteins: ELO1 is able to elongate C14 fatty acids (FAs) to C16 FAs, ELO2 can generate C24 FAs, and ELO3 plays essential roles in the conversion of C24 FAs to C26 FAs [59,60]. In our study, activation of GAL1 promoter by galactose successfully transcribed the target genes, and we detected the expression of GFP protein, which suggests that this expression system is effective (S7C–S7E Fig). GC–MS analysis of FAs in *S. cerevisiae* that contained pYES2-GFP (control) detected large amounts of C16:1, C16:0, C18:1, and C18:0 FAs, minor amounts of C20 to C24 and C28 FAs, and a larger amount of C26:0 FAs (S3 Table). Yeast with exogenous *BgElo12* did not produce any new FAs, but heterologous expression of *BgElo24* sharply increased the amount of C28:0 FA and generated a new component, C30:0 FA (Fig 4 A–C, S7H and S7I Fig). These results suggest that BgElo24 is capable of elongating endogenous yeast FAs to generate C28:0 and C30:0 FAs.

Considering that BgElo12 and BgElo24 may selectively elongate substrates with specific carbon chain lengths, we separately added C20:0, C22:0, C24:0, C26:0, and C28:0 FAs to the medium. Compared with the control, BgElo24 did not produce any new FAs when C20:0, C22:0, C24:0, or C26:0 were added (S3 Table). However, when C28:0 FA was added, yeast with pYES2-BgElo12 generated C30:0 FA, although in small amounts (Fig 4A'–4C', S3 Table). These results suggest that BgElo24 not only directly provides substrates (C28 and C30 fatty acyl-CoAs) for the biosynthesis of C27 and C29 *n*-alkanes, but also provides C28 fatty acyl-CoA for BgElo12 to elongate to C30 fatty acyl-CoA, which, in turn, generates C29 *n*-alkane. This might be the reason why in vivo RNAi of *BgElo12* slightly decreased C29 *n*-alkane and caused a dramatic increase in C27 *n*-alkane, whereas RNAi of *BgElo24* dramatically decreased C29 *n*-alkane.

In order to analyze the activity of BgElo12 and BgElo24 in elongating methyl-branched FAs, 2 representative substrates, 2-MeC16:0 and 14-MeC16:0 FAs, the potential substrates for 15-methyl HCs and 3-methyl HCs, respectively, were added to the medium. However, we found that these substrates could not be catalyzed by BgElo12 or BgElo24 (S7F–S7F" and S7G–S7G" Fig). We suspect that this might be caused by a shortcoming of the yeast FA elongation system. FA elongation requires an elongase and 3 other enzymes [29]. The last 3 yeast endogenous enzymes may not catalyze methyl-branched substrates. In conclusion, these results indicate that both BgElo12 and BgElo24 catalyzed the biosynthesis of VLCFAs in the yeast heterologous system, and BgElo24 was able to provide primary substrates for BgElo12. However, the activity of BgElo12 and BgElo24 in methyl-branched FA elongation needs further investigation.

## Female-enriched HCs are crucial for contact sex pheromone–based courtship performance

The largest single chromatographic HC peak in females contains 3,7-; 3,9-; and 3,11-DimeC29, which is much less represented in males. Because 3,11-DimeC29 has been shown to be the precursor for the contact sex pheromone 3,11-DimeC29-2-one (C29 methyl ketone) in *B. germanica* [48], we suspected that the female-specific HC profile may be important for maintaining a high level of contact sex pheromone. We first monitored the pattern of 3,11-DimeC29-2-one accumulation in the first gonotrophic cycle. The contact sex pheromone

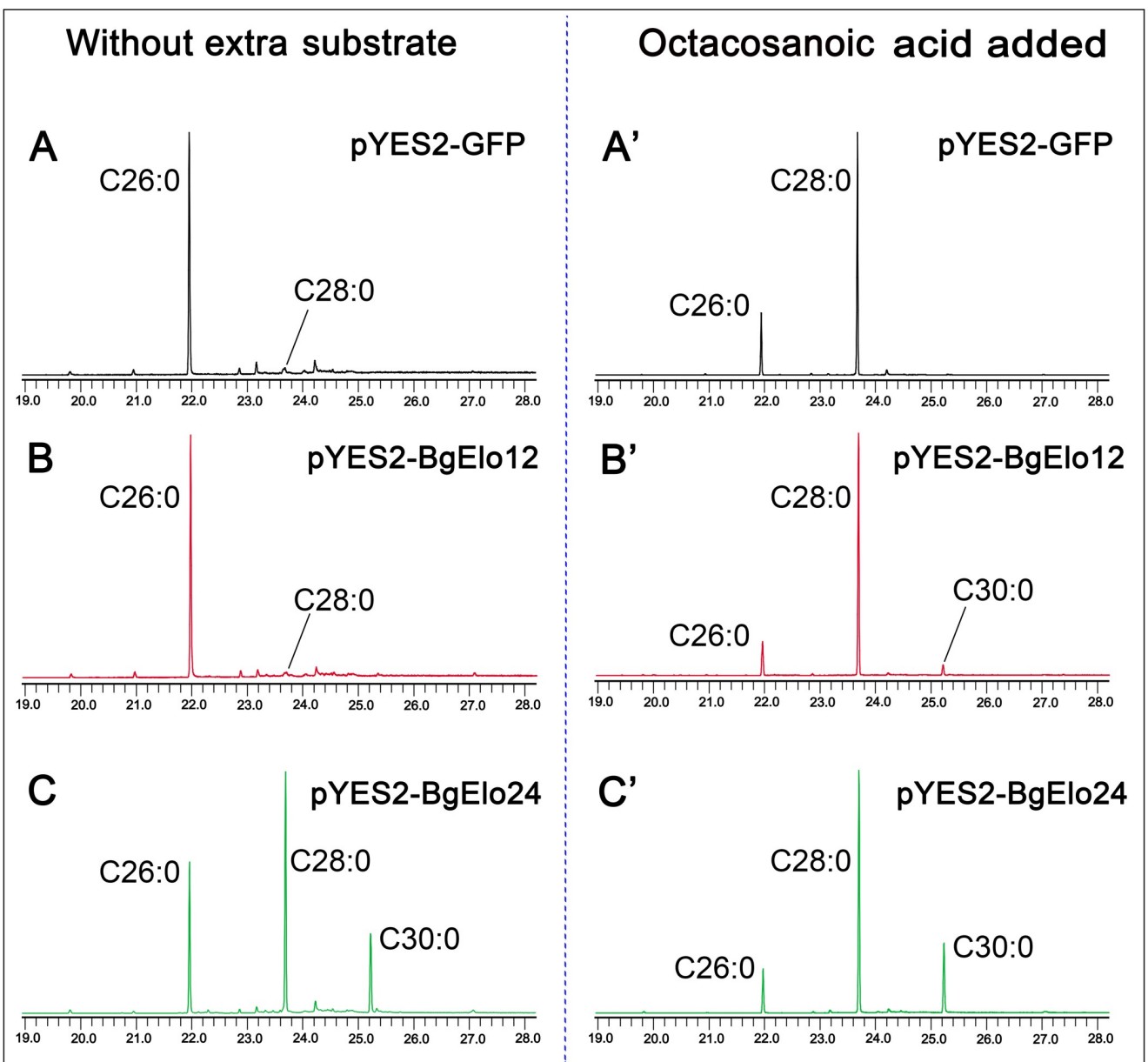

**Fig 4. Yeast expression and substrate catalysis of BgElo12 and BgElo24.** Representative gas chromatogram of FAMEs of yeast transformed with pYES2-GFP (**A**), pYES2-BgElo12 (**B**), and pYES2-BgElo24 (**C**) without addition of extra substrates. After adding octacosanoic acid (C28:0), the corresponding chromatograms are shown at the right for pYES2-GFP (**A'**), pYES2-BgElo12 (**B'**), and pYES2-BgElo24 (**C'**). The chromatograms are truncated, and only chromatographic peaks of interest are indicated. The FAMEs are labeled by their corresponding saturated FAs. Complete chromatograms are shown in S7 Fig. The data underlying this figure are included in S3 Table. FA, fatty acid; FAME, fatty acid methyl ester.

showed a stable low level at the early adult stage (AD0 to AD2), its accumulation started at AD3, quickly increased from AD5 to AD7 (Fig 5A), but decreased at AD8 when most females oviposited. Analysis of the influence of *BgElo12*-RNAi on C29 methyl ketone was performed at AD6. Female cockroaches were subjected to 3 consecutive dsRNA injections before being subjected to lipid analysis: the first one at early fifth instar, the second at early sixth instar, and the third at AD1. We found that repression of *BgElo12* reduced the C29 methyl ketone by more than 75% in both cuticular and internal extractions (Fig 5B–5D).

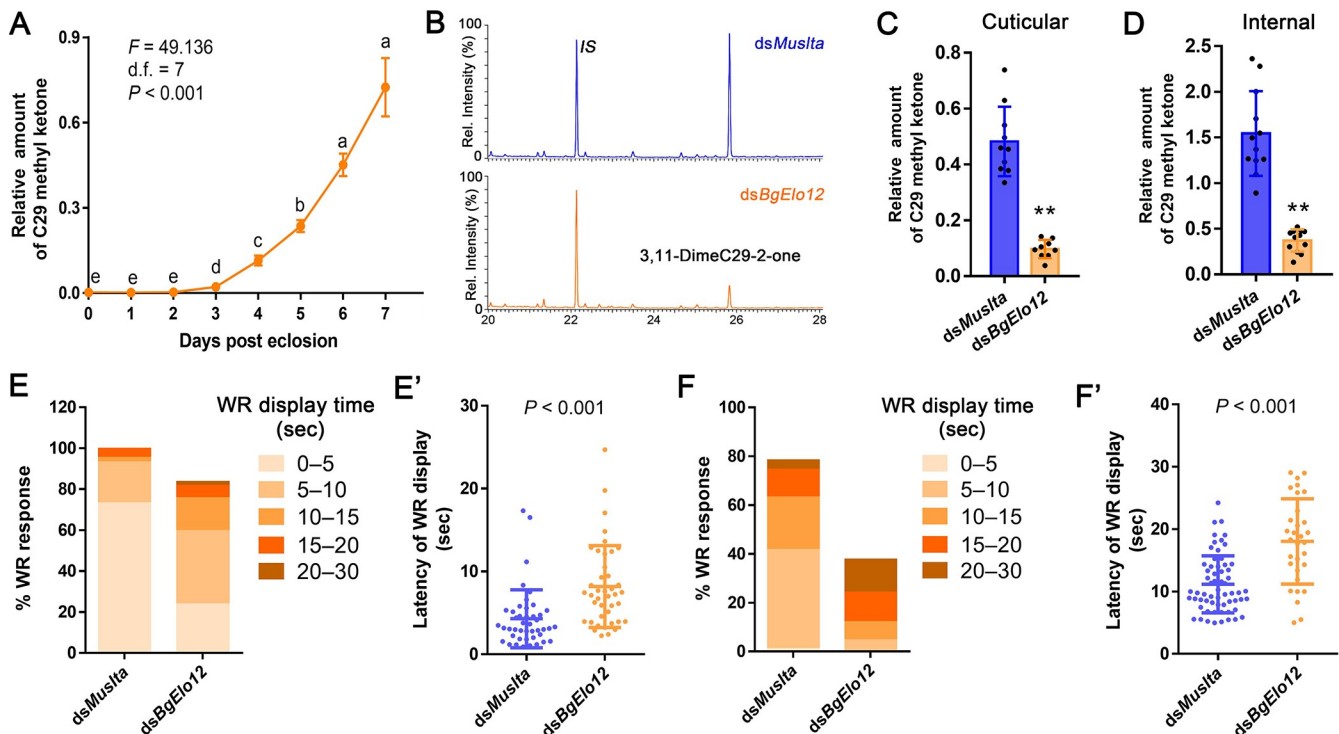

**Fig 5. *BgElo12*-RNAi effects on contact sex pheromone biosynthesis in females and male courtship performance. (A)** The pattern of contact sex pheromone (3,11-DimeC29-2-one) accumulation during female sexual maturation. Data are shown as mean ± SEM. Different letters indicate significant differences between groups using Welch ANOVA (Games–Howell multiple comparisons test, *P* < 0.05). **(B)** Representative chromatogram showing lower amounts of sex pheromone after *BgElo12*-RNAi. *IS* is the internal standard 14-heptacosanone. Relative cuticular **(C)** and internal **(D)** amounts of 3,11-DimeC29-2-one (C29 methyl ketone) after *BgElo12*-RNAi. Data are shown as mean ± SD, and each replicate is shown as a dot. **P* < 0.01, 2-tailed Student *t* test; *n* = 9–12. Influence of *BgElo12*-RNAi on the ability of AD5 **(E–E')** and AD3 **(F–F')** female antennae to elicit courtship in males. The percentage of males that responded with a WR behavior in response to contact with female antenna were determined for 44 (ds*Muslta*) and 49 (ds*BgElo12*) female antennae in **(E)** and 79 (ds*Muslta*) and 82 (ds*BgElo12*) antennae in **(F)**. Each female antenna was tested only once with a single male. The proportion of WR display over different time periods (latency, seconds) is shown in progressively darker colors. **(E')** and **(F')** Average latency of WR display calculated from those antennae that successfully activated a courtship behavior of males within 30 seconds. Data are shown as mean ± SD; each dot represents a datum calculated from one antenna. *P* values were determined by 2-tailed Student *t* test. The data underlying this figure are included in S1 Data. AD, adult day; RNAi, RNA interference; WR, wing raising.

We next determined whether the *BgElo12*-RNAi repression of pheromone production in females affected male courtship behavior. When the female contact sex pheromone is detected by a male, it displays a characteristic male courtship behavior, including wing raising (WR) [50]. When an antenna of AD5 female was used as stimulus, nearly all males responded. RNAi of *BgElo12* resulted in only a slight decline in WR rate (about 20%), but the latency of WR was significantly increased (Fig 5E and 5E'). For using the antenna of AD3 females, we found that males responded to 80% of the control antennae with the WR display, but less than 40% of the antennae from *BgElo12*-RNAi females elicited WR in males. The average latency of WR toward antennae of *BgElo12*-RNAi females was 18.03 seconds, while the control antennae elicited WR in 11.05 seconds (Fig 5F and 5F'). These results indicate that female-enriched HCs are advantageous for contact sex pheromone biosynthesis, especially at the early sexual maturation stage.

## Sex differentiation genes modulate *BgElo12*

The female-enriched HC composition is tightly associated with a higher expression of *BgElo12* in females, but the regulators that govern sexually dimorphic expression of *BgElo12* in females and males are unknown. Wexler and colleagues reported that RNAi of the sex determination

gene *BgTra* in females converts its CHC profile to a male-like profile [52]. Taken together with our results, it would appear that the sex differentiation pathway might be involved in regulating *BgElo12* expression. We first confirmed the function of *BgTra* and *BgDsx* in sex-specific development via RNAi. For each gene, we used 2 unique RNAi targets within a conserved sequence region for multiple isoforms. RNAi of *BgTra* and *BgDsx* in females or males showed similar phenotypes as described by Wexler and colleagues [52]. *BgTra* is only functional in females, while *BgDsx* only works in males (S8 Fig). We next studied the expression of *BgTra* and *BgDsx* in females and males using quantitative PCR (qPCR) primers that target all isoforms of each gene. The expression of both *BgTra* and *BgDsx* increased after eclosion, but *BgTra* transcript levels tended to be stable from AD4 to AD6 (Fig 6A and 6B). RNAi of *BgTra*

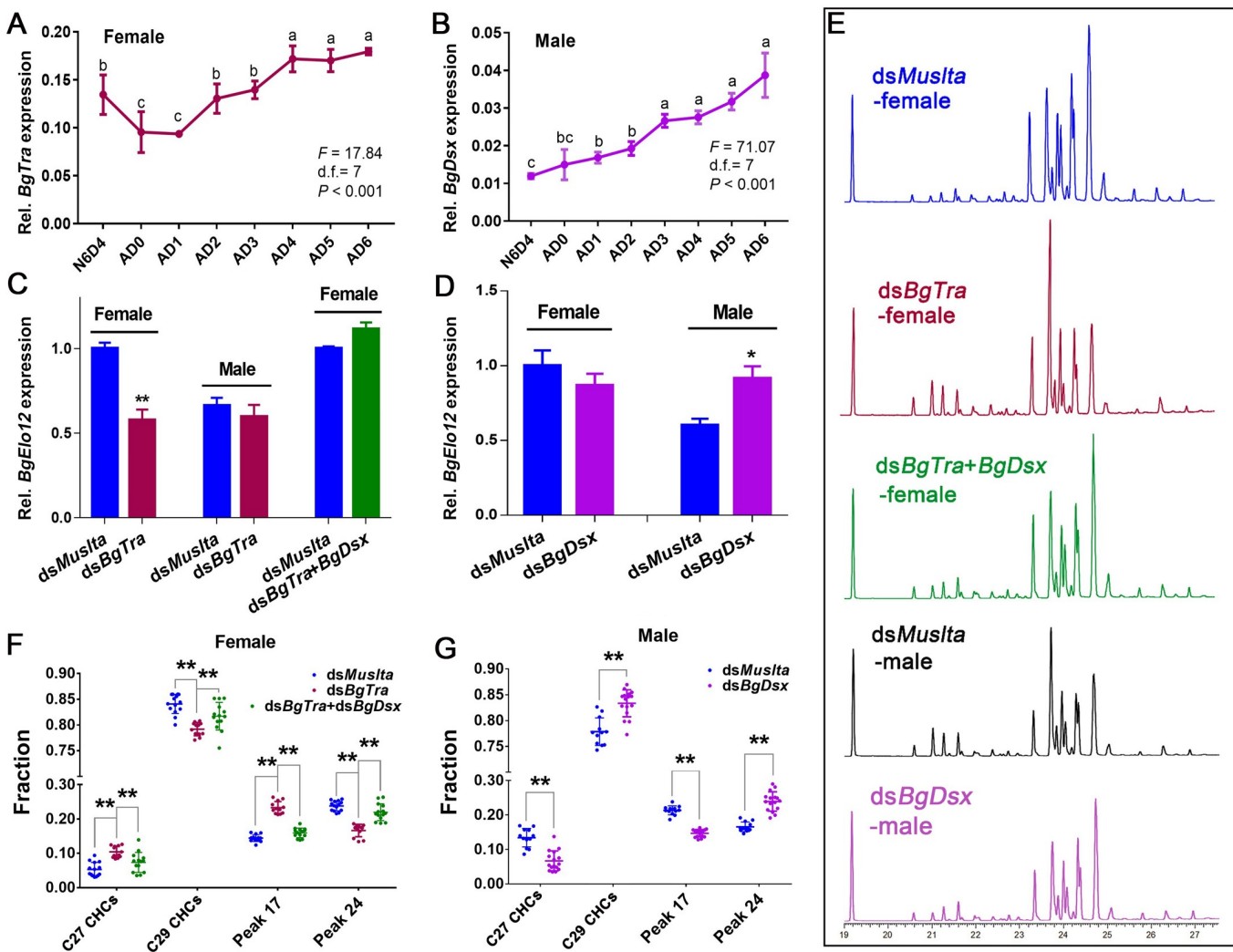

**Fig 6. Regulation of *BgElo12* and SDHC profiles by sex differentiation genes.** Temporal patterns of *BgTra* expression in females (**A**) and *BgDsx* expression in males (**B**). Data are shown as mean ± SEM and calculated from 4 replicates (2–3 cockroaches/replicate). N6D4 is sixth-instar nymph on day 4, AD0–AD6 represent adult days 0 to 6. Different letters indicate significant differences between the groups (ANOVA, Fisher LSD, *P* < 0.05). (**C, D**) Regulation of *BgElo12* expression by *BgTra* and *BgDsx*. Data are shown as mean calculated from 4 replicates (2–3 cockroaches/replicate) ± SEM; *\*P* < 0.05, *\*\*P* < 0.01, 2-tailed Student *t* test. (**E**) Gas chromatograms of CHCs after repressing different sex differentiation genes. (**F**) Regulation of the proportions of representative CHCs by *BgTra* in females. Data are presented as mean ± SD, *\*\*P* < 0.01; differences between 2 groups were determined by 2-tailed Student *t* test, differences among 3 groups were determined by 1-way ANOVA and LSD multiple comparisons; *n* = 14 for *dsMuslta*, 12 for ds*BgTra*, and 14 for ds*BgTra*+*BgDsx*. (**G**) The proportion change of representative CHCs after *BgDsx*-RNAi in males. Data are shown as mean ± SD; *\*\*P* < 0.01; 2-tailed Student *t* test, *n* = 12 for *dsMuslta* and 16 for ds*BgDsx*. The data underlying this figure are included in S1 Data. AD, adult day; CHC, cuticular hydrocarbon; LSD, least significant difference; RNAi, RNA interference; SDHC, sexually dimorphic hydrocarbon.

in females significantly decreased the mRNA level of *BgElo12*, but *BgElo12* expression was unaffected in males (Fig 6C). Notably, the down-regulation of *BgElo12* by ds*BgTra* injection in females could be rescued by co-injection of ds*BgTra* and ds*BgDsx* (Fig 6C), but ds*BgDsx* alone was not functional in regulating *BgElo12* expression in females (Fig 6D). Conversely, in males, knockdown of *BgDsx* alone increased the expression of *BgElo12* (Fig 6D). As RNAi of *BgTra* in females generated the male type *BgDsx* (*BgDsx*$^M$) [52], we suspect that the down-regulation of *BgElo12* in females after RNAi of *BgTra* was caused by the generation of *BgDsx*$^M$. This hypothesis was supported by co-injection of ds*BgTra* and ds*BgDsx* and a dual-luciferase reporter assay that BgDsx$^M$ could directly modulate the expression of *BgElo12* by interaction with its upstream sequence region (S9 Fig). Further, we found that RNAi of neither *BgTra* in females nor *BgDsx* in males could affect the expression of *BgElo24* (S10 Fig), suggesting that *BgElo24* is not regulated by these 2 sex differentiation genes.

We further analyzed the effects of *BgTra* and *BgDsx* on HC profiles. Knockdown of *BgTra* in females masculinized the CHC profile and increased the male-enriched CHC components; however, this change disappeared by co-injection of ds*BgTra* and ds*BgDsx* (Figs 6E and 6F, S11A). Similarly, repression of *BgDsx* in males feminized the CHC profile and increased the female-enriched components (Figs 6E and 6G, S11B). In order to verify that the sex differentiation genes affected CHC profiles by regulating the generation of HCs but not their selective transport and deposition to the cuticle, we analyzed the internal HCs. RNAi of both *BgTra* in females and *BgDsx* in males caused an intersexual conversion of internal HC profiles, as seen in the CHCs (S12A–S12D Fig). Overall, our results indicate that the sexual dimorphism of HCs in *B. germanica* is primarily determined by *BgElo12*, and sex determination pathway genes are the critical regulators that control the asymmetric expression of *BgElo12* between males and females.

## Discussion

Our study reveals a novel molecular mechanism responsible for the formation of SDHCs in *B. germanica*. The CHC profiles in insects are regulated by complex biosynthetic and transport pathways, involving multiple gene families. We demonstrated that the fatty acid elongation step is responsible for sexual dimorphism of CHCs in *B. germanica*, We identified that both *BgElo12* and *BgElo24* were participated in HC biosynthesis, but only the female-enriched *BgElo12* is the core gene that encodes for the elongase involved in generating more female-enriched HCs, and the asymmetric expression of *BgElo12* between the sexes is modulated by sex differentiation genes: *BgDsx*$^M$ represses the expression of *BgElo12* in males, while *BgTra* removes this repression in females (Fig 7). Because a female-enriched HC serves as a precursor to a female contact sex pheromone, we also revealed the prominence of *BgElo12* in sexual behavior.

### Fatty acid chain elongation is a key step in the regulation of SDHCs in *B. germanica*

The diversity of HCs in insects is reflected in the HC carbon chain lengths, their degree of saturation, and the number and positions of methyl groups [39]. HCs in *B. germanica* are composed of only *n*-alkanes and methyl-branched alkanes, and the fatty acid biosynthesis gene that governs the incorporation of methyl groups in the aliphatic chain showed no role in generating the sexual dimorphism of HCs in *B. germanica* [55]. In this study, we found that sexually mature female cockroaches contain relatively higher amounts of C29 HCs than males, whereas male cockroaches had more C27 HCs, suggesting that chain length is an important factor in the dimorphism of HCs.

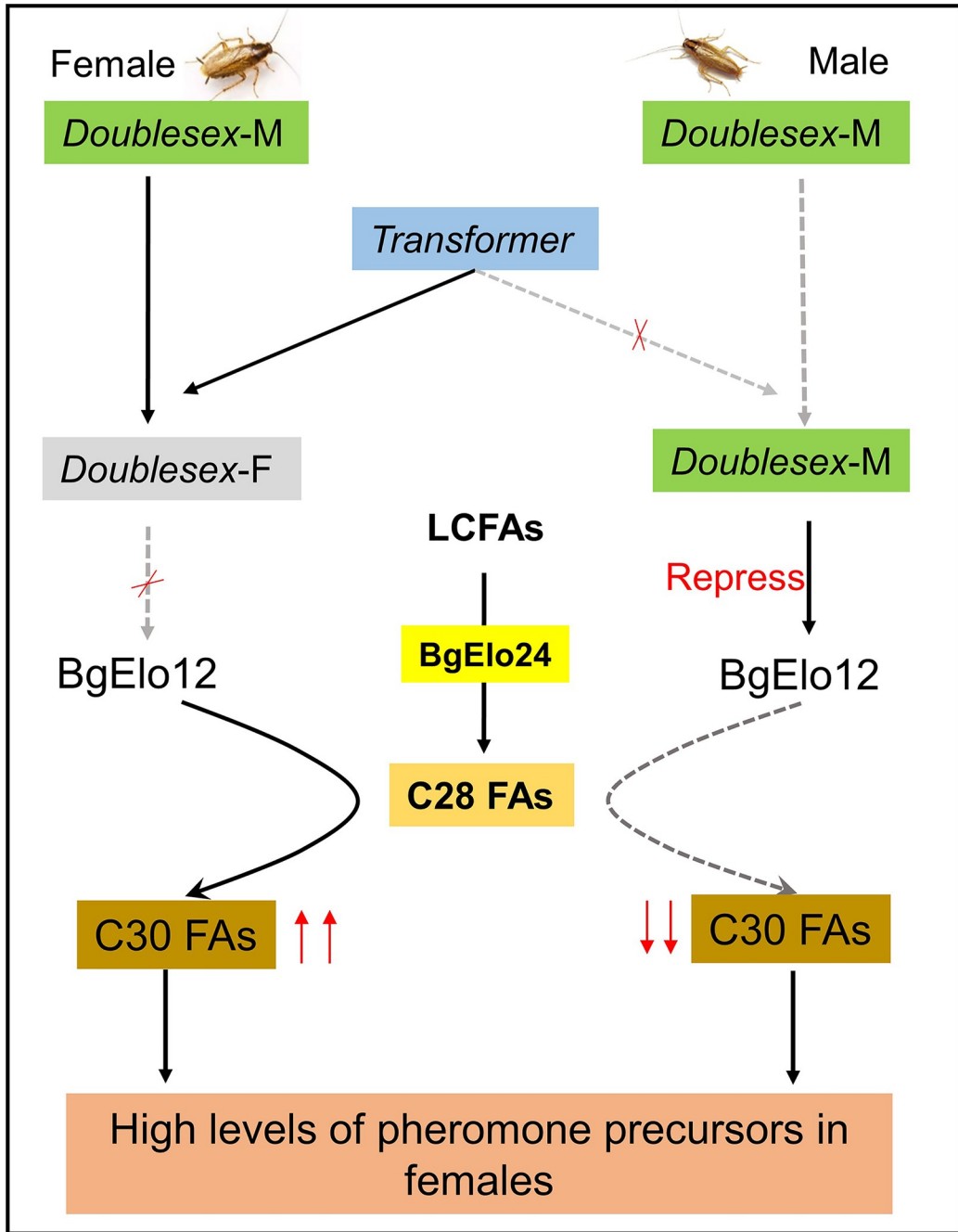

**Fig 7. Model for *BgElo12* in generation SDHCs in *B. germanica*.** Both *BgElo12* and *BgElo24* participated in HC biosynthesis, but *BgElo24* is a basic fatty acid elongase that is able to catalyze a wide range of substrates and provides C28 FAs for *BgElo12* to generate specific C30 FAs. The male-specific *Doublesex* (*Doublesex-M*) is able to repress the transcription of *BgElo12* in males. However, *Transformer* only functions in females and can splice *Doublesex-M* to the nonfunctional female type *Doublesex* (*Doublesex-F*); thus, *BgElo12* is highly expressed in females and generates more C30 FAs. A high level of C30 FAs lastly produces more contact sex pheromone precursors in females. FA, fatty acid; HC, hydrocarbon; LCFA, long-chain fatty acid; SDHC, sexually dimorphic hydrocarbon.

Studies of the genetic bases of HC biosynthesis in insects have demonstrated that chain lengths are determined by the fatty acid elongation process: The rate-limiting enzyme elongase determines the chain lengths of VLCFA products and produces the final HCs with different

chain lengths [30]. In our study, an RNAi screen identified that *BgElo12* and *BgElo24* were involved in HC biosynthesis. Expression of both genes in yeast demonstrated that both BgElo12 and BgElo24 displayed functions similar to ELOVL4 in mammals, which showed a special function in the biosynthesis of VLCFAs with chain lengths greater than C28 [61]. *BgElo24*, however, is a more fundamental elongase that seems able to catalyze a wide range of substrates to generate FAs of various chain lengths and can also provide primary substrates for BgElo12 to produce C30 FAs. Notably, only *BgElo12* had higher expression in females, which was consistent with the higher amounts of C29 HCs in females. Knockdown of *BgElo12* in females caused a dramatic decrease of the female-enriched peak 24 (3,7-; 3,9-; and 3,11-DimeC29), but it did not affect the male-enriched peak 17 (9-; 11-; 13-; and 15-MeC29), which generated a male-like HC profile; however, RNAi of *BgElo24* unselectively down-regulated all C28 to C30 HCs, indicating that only *BgElo12* was involved in the formation of SDHCs in *B. germanica*. Lastly, we found both RNAi of *BgElo12* and *BgElo24* increased some C27 HCs, indicating that there are some other *BgElos* involved in HC production. The identification of these genes will be arduous, because the C27 compounds occur in small amounts, and multiple BgElos may catalyze the synthesis of the same HC independently. It is worth noting in this regard that *Blattella asahinai*, a sister species of *B. germanica* produces almost no C27 compounds [62]. These 2 species can hybridize, potentially offering a resource for the genetic regulation of C27 HCs.

In *B. germanica*, as in other insects, HCs are transported through the hemolymph and selectively incorporated into or deposited on various tissues, including the cuticle, ovaries, and specialized pheromone glands [47,63,64]. In our study, analysis of the influence of *BgElo12*-RNAi on internal HCs demonstrated that the female- and male-specific CHC profiles were not caused by selective transport, but rather by differences in de novo HC biosynthesis between the sexes, regulated by *BgElo12*. Also, the total amount of internal HCs was significantly decreased in *BgTra*-RNAi females, while it was increased in *BgDsx*-RNAi males (S12E and S12F Fig). These results suggest that there is an unusual transport of HCs after repression of *BgTar* or *BgDsx*. We suspect that the unusual transport of HCs may be caused by changes in the capacity to store internal HCs, as large amounts of internal HCs are shunted to the ovaries [45,65], and sex differentiation genes are the key regulators of normal ovary development in females and repression of ovary generation in males [8,52]. We conclude that although the transport of HCs from internal tissues to the cuticle was affected after repressing *BgDsx* or *BgTra*, the effect was on overall HC transport, with no apparent selective transport on specific HCs. The results suggest that the influences of sex differentiation genes on SDHCs are not caused by selective transport, but may largely relay on the regulation of *BgElo12* expression. Elongases have been shown to participate in HC production in several other insect species including *D. melanogaster*, *Nilaparvata lugens*, and *Locusta migratoria* [33,66,67]. The elongase gene *eloF* was also shown to be specifically expressed in females and may be regulated by the sex differentiation gene *Transformer*, but the specific details are not very clear [33]. Nevertheless, the roles of elongase genes in sexual dimorphism of HCs in other insects have not been described.

## *BgElo12* and *BgDsx*$^M$ are key nodes connecting the HC synthesis and sex differentiation pathways

In this study, we first demonstrated that *BgElo12* is the key regulator in the HC biosynthesis pathway, responsible for the differences in the HC profiles between females and males. We next explored the upstream regulators that modulate the sexually dimorphic expression of *BgElo12*. The molecular genetic switches that determine which sex determination pathway is

followed by males and females are highly variable in animals. The *doublesex/mab-3 related* (*Dmrt*) family of transcription factors includes conserved developmental regulators in the sex differentiation pathway, governing the fate of sexually dimorphic traits in animals [8]. *Dsx* in arthropods, which is related to *Dmrt*, works through sex-specific splice variants that are controlled by *Tra* in many insects. Sex-specific *Dsx* isoforms promote sexual differentiation by modulating diverse downstream genes; thus, the *Dsx* gene is regarded as a central nexus in sexual differentiation [68,69]. Previous work reported that knockdown of *BgTra* converted the female cockroach CHC profiles to male-like profiles [52]. In our study, RNAi of *BgTra* in female cockroaches indeed down-regulated the expression of *BgElo12*, but the effect of *BgTra*-RNAi on *BgElo12* expression could be recovered by co-injection of ds*BgTra* and ds*BgDsx*, and RNAi of *BgDsx* in males up-regulated *BgElo12* transcripts. These results suggest that $BgDsx^M$ can repress the transcription of *BgElo12* in males, and the down-regulation of *BgElo12* by repressing *BgTra* was caused by the conversion of $BgDsx^F$ to $BgDsx^M$, as *BgTra* is able to regulate the splicing of $BgDsx^M$ to $BgDsx^F$ [52]. The regulation of *BgElo12* by *BgDsx* connects the sex differentiation pathway with the HC biosynthesis pathway, and, therefore, enables the sexual dimorphism of HCs in *B. germanica*.

Although the dual-luciferase reporter gene assay suggested that $BgDsx^M$ can directly regulate the transcript of *BgElo12*, there may be other indirect regulatory pathways. Several candidate factors have been shown to regulate the HC profiles in various insects (e.g., *D. melanogaster* and *Musca domestica*), including ecdysone, juvenile hormone (JH), biogenic amines, and the insulin signaling pathway [26,70–77]. It has been reported that ecdysone is able to regulate the fatty acyl-CoA elongation step and shift the chain lengths of HCs in *M. domestica* [26]. Ecdysone is mainly produced by the prothoracic gland, but, in some insects, also by the ovaries [78,79], and the development of the ovaries is under the regulation of the sex differentiation pathway. Moreover, endocrine signals like JH and insulin are also regulated by the sex differentiation pathway [80]. Therefore, these signals may be potential mediators that complete the regulatory network between *BgDsx* and *BgElo12*. However, more detailed investigations are needed to thoroughly elucidate the differential regulation of HC production in both females and males.

*Dsx* also regulates the HC biosynthesis pathway in *D. melanogaster*; $Dsx^F$ specifically activates the transcription of *desatF* and generates pheromonal dialkenes [42]. However, $BgDsx^M$ suppresses the generation of female traits in male cockroaches, while *BgTra* removes the inhibitory effect of $BgDsx^M$ in females [52]. Thus, $BgDsx^M$ represses the expression of *BgElo12*, and, therefore, it represses the generation of female-enriched HCs in males, and *BgTra* is crucial in maintaining female-enriched HCs, especially for the contact sex pheromone precursors. However, *BgTra* may also affect contact sex pheromone biosynthesis in female cockroaches through additional pathways.

## Biological significance of the sexual dimorphism of HCs

Sexually dimorphic traits are generated in females and males in response to intra- and intersexual selection, but pleiotropic traits are also subject to natural selection, especially when they are also shaped by and adapted to environmental stresses [3,81]. Most moth sex pheromones are C12 to C18 aldehydes, alcohols, and acetate esters, derived from fatty acids in specialized pheromone glands. These pheromones appear to have no other function beyond attracting the opposite sex. In contrast, CHC pheromones appear to serve both in sexual communication and in waterproofing of the cuticle. The female-specific contact sex pheromone of *B. germanica* is clearly subject to both natural and sexual selection because it is derived from a prominent CHC, which also serves as a waterproofing component of the CHC profile and is maternally invested in offspring [82]. Thus, maintenance of HC contact sex pheromone

precursors in female cockroaches is adaptive in both reproductive success and survival in an arid environment. The quality of insect pheromones is considered an honest indicator of fitness potential [73]. This assertion may be particularly pertinent in *B. germanica*, where the pheromone and its HC precursor serve in sexual communication, resilience to environmental stressors, and in maternal investment in eggs.

In systems that use CHCs in sexual communication, it is common for the CHC profiles to contain female- and male-specific components, as is evident in *Drosophila*. In *B. germanica*, as well, male cockroaches have a unique CHC profile, especially enriched in C27 components with 9-; 11-; 13-; and 15-MeC29 being particularly prominent. The male-specific HC profile is generated during sexual maturation, suggesting that it may function in sexual communication. It is possible that the male-enriched 9-; 11-; 13-; and 15-MeC29 may function as a sex pheromone in 3 related contexts: (a) it may distinguish males and females within cockroach aggregations; (b) it may signal "maleness" and male quality to females; and (c) it may function in male–male recognition, contests, and competition for access to females. Moreover, it is possible that male-specific P450s may catalyze the oxidation of these male-enriched HCs to homologous methyl ketones, as in females, and, in turn, serve these functions. However, more bioassays are required to analyze the biological significance of male-specific CHC profiles.

## Materials and methods

### Insect rearing

The German cockroach *B. germanica* originated from a laboratory strain collected in the 1970s. The cockroaches were maintained in aquaria at 30 ± 1˚C with a relative humidity (RH) of approximately 50% under 12:12 hour light–dark photoperiod regime and fed rat chow and tap water. Newly hatched cockroaches were separated and reared in new containers. Early-stage (day 1 or 2) fourth- and fifth-instar nymphs were separated and used in dsRNA injection; newly emerged adults were collected and reared in plastic jars for experiments.

### Preparation of HCs and methyl ketones

*B. germanica* cuticular lipids were extracted as described [55]. Individual adult female cockroaches were sacrificed by freezing at −20˚C, thawed at room temperature, the cuticle surface-extracted in 1 mL of hexane twice, and finally rinsed in 1 mL of hexane. *n*-Hexacosane (15 μg) or 14-heptacosanone (0.5 μg) were added as internal standards. The extracts were combined and reduced to approximately 300 μL with a nitrogen flow and loaded onto a Pasteur pipette silica gel mini-column, as previously described [55]. The CHCs were eluted with 8 mL of hexane, and the contact sex pheromone fraction was subsequently eluted with 8 mL of 3% ethyl ether in hexane. Internal lipids were extracted from the cockroach following a procedure described by Fan and colleagues [83]. Each surface-extracted cockroach was homogenized in a solution of hexane–methanol–ddH2O (2:1:1 mL), and 30 μg of *n*-hexacosane and 1 μg of 14-heptacosanone were added for quantification of internal HCs and methyl ketones, respectively. The homogenate was vigorously vortexed and centrifuged at 2,500 g for 10 minutes. The supernatant hexane phase was collected, and the extraction was repeated using *n*-hexane. Separation of HCs and methyl ketones was performed using column chromatography, as described above.

### GC–MS analysis

Lipid analysis was performed with a TRACE 1310 GC–ISQ single quadrupole MS (Thermo Fisher Scientific, Waltham, Massachusetts, United States of America). In brief, lipids were separated on a DB-5MS capillary column (30-m length, 0.25-mm ID, and 0.25-μm film thickness;

Agilent Technologies, Santa Clara, California, USA). The oven started at 60˚C and kept for 2 minutes, heated to 160˚C for HCs and methyl ketones, or 220˚C for fatty acid methyl esters (FAMEs) at a rate of 30˚C/min, then increased at 3˚C/min up to 250˚C, followed by 10˚C/min up to 320˚C and held for 5 minutes. Electron ionization mode (70 eV) was used, and the MS scan range was 45 to 650 m/z at a rate of 5 scans/s. Identification of HC compounds was performed according to Jurenka and colleagues [54], and peak area determination was performed with an Xcalibur 2.2 workstation.

## RNA isolation and real-time quantitative PCR (RT-qPCR)

Total RNA was isolated with RNAiso Plus Reagents (Takara, Dalian, Liaoning, China) according to the manufacturer's instructions. cDNA was reverse-transcribed from 800 ng of total RNA using the PrimeScrip RT reagent Kit with gDNA Eraser (Takara). Gene-specific primers with appropriate amplification efficiency (0.95 to 1.05) were screened by a cDNA dilution series (S4 Table). Quantification of gene expression level was performed with TB Green Premix Ex Taq Tli RNase H Plus (Takara) on a LightCycler 480 system (Roche, Basle, Switzerland). Target genes expression was normalized by the commonly used housekeeping gene *actin5c* (GenBank: AJ862721.1) and calculated using the $2^{-\Delta\Delta Ct}$ method. Each treatment contained 4 biological replicates and technical triplicates.

## Identification of *BgElo* gene family members

Both BLASTN and BLASTP were used to search *BgElo* genes in *B. germanica* genome data [57] and our own full-length transcriptome data (NCBI accessions: SRR9143014 and SRR9143013) using the homologous genes of *Elongase* from *D. melanogaster* as query sequences. Candidate *BgElo* genes were amplified with the PrimeSTAR GXL DNA Polymerase reagent (Takara) (primers are listed in S4 Table), and the amplified fragments were inserted into the pMD 19-T Vector (Takara) and re-sequenced. Candidate *BgElo* genes were then translated and submitted to SMART online tools (http://smart.embl-heidelberg.de) to analyze the conserved structures; only genes with the typical ELO domain were confirmed as *BgElo* genes. The putative *BgElo* mRNA sequences were mapped to the genomic data (GenBank: PYGN00000000.1) using a local BLASTN tool, and the intron–exon structure was analyzed based on the GT–AG rule. The conservative motifs in *BgElos* were analyzed by sequence alignment with DNAMAN 9.0 software.

## Expression profile analysis

In order to screen the potential *BgElo* genes involved in HC biosynthesis, the expression levels of different *BgElo* genes were quantified in the fat body and abdominal integument, where HCs or their precursors were generated. In addition, other tissues including the head, thorax, gut, legs, ovaries, Malpighian tubules, ejaculatory duct (from 2-day-old males), and colleterial glands were dissected from 2-day-old females and were used to analyze the expression profiles of *BgElo12* and *BgElo24* among different tissues. In order to study the time course of *BgElo12*, *BgElo24*, *BgDTra*, and *BgDsx* transcript levels during sexual maturation, a representative nymphal stage (4-day-old sixth-instar nymph, N6D4) and AD0 to AD6 (0-day-old to 6-day-old adult) females and males were collected. Total RNA was extracted from various tissues or intact cockroaches, and the expression profiles of different genes were studied via RT-qPCR.

## Transcript knockdown via RNAi

Gene-specific target sequences as well as a heterologous fragment from *Mus musculus* (*Muslta*) used for dsRNA synthesis were amplified and cloned into pMD 19-T Vector (Takara).

Templates used for single-stranded RNA were amplified with primers that incorporated with the T7 promoter sequence (S4 Table). Different kinds of dsRNA were subsequently generated with the T7 RiboMAX Express RNAi System (Promega, Madison, Wisconsin, USA). Delivery of dsRNA was performed with a Nanoject II micro-injector (Drummond Scientific, Broomall, PA, USA) for fourth-instar cockroaches and microliter syringes for fifth-, sixth-instar, and adult cockroaches. For RNAi screen of *BgElo* genes in HC biogenesis, a double-injection strategy was employed: The first injection was performed at the early fifth-instar (1- or 2-day-old fifth-instar, N5D1-N5D2) with a dosage of 3 μg in 2 μL; the second injection was performed 1 week later with a dosage of 4 μg in 2 μL. For confirming the function of *BgElo12* in contact sex pheromone biosynthesis, a third injection with 4 μg of dsRNA in 2 μL was carried out on (AD1, and methyl ketones were extracted at AD6. Knockdown of sex determination genes was accomplished with 3 dsRNA injections, the first at early fourth instar (N4D1 to N4D2), the second at early fifth instar (N5D1-N5D2), and the last at early sixth instar (N6D1 to N6D2). The fourth-instar cockroaches were injected with approximately 0.5 μg of dsRNA in approximately 0.2 μL; the fifth- and sixth-instar cockroaches were injected with 1 μg of dsRNA in 2 μL. In order to verify the function of *BgElo12* and *BgElo24* in HC biosynthesis and sex determination genes in modulating *BgElo12* mRNA levels or HC profiles, 2 nonoverlapping gene-specific targets were designed and used in this study; as there are some isoforms of *BgDsx* and *BgTra*, the RNAi targets were designed within the common sequence region. After dsRNA injection, intact AD2 cockroaches were used for RNAi efficiency analysis.

## Heterologous expression and fatty acid analysis

Heterologous expression was performed as described [84]. Complete coding sequences (CDSs) of *BgElo12*, *BgElo24*, or control (*GFP*) were amplified with PrimeSTAR HS DNA Polymerase (Takara) using gene-specific primers (S4 Table) that contain the restriction enzyme sites (*KpnI* and *BamHI* for *BgElo12* and *BgElo24*; *BmHI* and *EcoRI* for *GFP*) and the yeast consensus sequence (TACACA) following the restriction enzyme sites (only for forward primers). The amplified *BgElo12*, *BgElo24*, or *GFP* CDS fragments were ligated into the linearized pYES2 shuttle plasmid (Thermo Fisher Scientific) and verified by sequencing. The recombinant plasmids were transformed into INVSc1 *S. cerevisiae* (Thermo Fisher Scientific) using the PEG-LiAc method and streaked onto *S. cerevisiae* minimal medium minus uracil (SC–uracil) plates to select transformants; single colonies were inoculated in SC–uracil medium with 2% glucose. After culturing at 30°C for 24 hours, the yeast was collected and diluted to an OD600 of 0.4 with SC–uracil medium containing 1% raffinose and 2% galactose and further cultured at 30°C until they reached an OD600 of 0.8. At this point, transcription of exogenous genes was examined by real-time PCR (RT-PCR). Substrates of C20 (0.5 mM), C22 (1 mM), C24 (1 mM), C26 (1 mM), C28 (1 mM), 2-methylhexadecanoic acid (0.5 mM), and 14-methylhexadecanoic acid (0.5 mM) were separately added into the cultures with an extra 1% of tergitol type Nonidet P-40. All the substrates were purchased from Sigma-Aldrich (Louis, Missouri, USA) or TCI (Shanghai, China).

After 48 hours, yeast cells were harvested by centrifugation at 500 g for 5 minutes and washed thrice with Hanks' balanced salt solution for fatty acid derivatization and analysis. Pellets were dried under a steam of nitrogen, and 2 mL of 1% (v/v) $H_2SO_4$ in methanol was added, and the mixture was vortexed and incubated at 80°C in a $N_2$ atmosphere for 2 hours [85]. After that, 1 ml of saturated sodium chloride solution was added into the mixture, and FAMEs were extracted with 1 mL of hexane 3 times. The FAME extracts were concentrated and subjected to GC–MS analysis as described above. Different FAMEs were identified by comparison to FAME standards (purchased from Sigma-Aldrich) and their mass spectra.

## Desiccation bioassay

The capacity of *BgElo12* and *BgElo24* to contribute to water retention was assessed by a desiccation bioassay. Drying bottles were prepared by putting approximately 120 g of packed fresh silica gel into an approximately 900-mL sealed plastic bottle. The RH inside the bottle dropped to 5% within 2 hours, which was monitored by HOBO Pro v2 (Onset, Bourne, Massachusetts, USA). AD2 females were injected with ds*BgElo12*, ds*BgELo24*, and ds*Muslta* and separately caged in the desiccation bottles at 30˚C, supplied with approximately 1 g of dry food, but no water. Survival was recorded every 8 hours until all the cockroaches died. About 100 cockroaches were used for each treatment.

## Courtship behavioral study

Courtship behavior was tested as described [86,87]. We firstly used antennae from AD5 females; each antenna was attached to the tip of a glass Pasteur pipette with paraffin, and the antenna was used immediately to test the responses of AD13 to AD15 males that were separated from females since eclosion. The test antenna was used to touch the antennae of the male, and a positive response was recorded if the male cockroach turned its body and raised the wings to approximately 90 degrees within 30 seconds. A negative response was recorded if the test antenna failed to elicit a response in a male cockroach and this male then responded to a positive control antenna from a normal AD6 female. The WR latency was recorded according to Wada-Katsumata and Schal [86]; the latency of the WR display was timed from contact of the antennae to the initiation of the male WR display. All female antennae and male cockroaches were used only once; all tests were performed in the scotophase, and we avoided the first and last 2 hours of the scotophase. Bioassays were conducted under a dim red light to simulate a dark environment. However, as AD5 females accumulated a large amount of contact sex pheromone, whereas males can be fully activated by about 10 ng of 3,11-DimeC29-2-one applied onto an antenna [88], we repeated this experiment using AD3 females, which have less contact sex pheromone on the cuticle.

## Dual-luciferase reporter gene assay

The 5′ end of *BgElo12* was obtained by 5′ Rapid Amplification of cDNA Ends (RACE). The 5′ RACE cDNA library was prepared using Clontech SMARTer RACE 5'/3' Kit (Takara) according to the user manual with the gene-specific primer (S4 Table) and kit-provided Universal long primer. The amplified fragments were cloned into pRACE vector and sequenced. About a 2.7-kb sequence upstream of *BgElo12* was amplified and cloned into pGL3-basic vector, and the CDS sequences of *BgDsx^M* and GFP (control) were separately cloned into the expression vector pCDNA3.1. The HEK293T cells were cultured in a 24-well plate with 500 μL of Dulbecco's Modified Eagle Medium (DMEM) (Thermo Fisher Scientific) for 24 hours before transfection, and the restructured pGL3-basic vector (200 ng/well) was co-transfected with the expression vectors (200 ng/well) to HEK293T cells using Lipofectamine 3000 (Invitrogen, Carlsbad, California, USA). The pRL-TK that encoded a *Renilla* luciferase was also co-transfected as an internal control. The transfected cells were cultured at 37˚C for 36 hours and subjected to luciferase activity analysis using the Dual-Glo Luciferase Assay System (Promega).

## Statistics

Data were statistically analyzed using SPSS 23 and presented as mean ± SEM or mean ± SD. Two-tailed Student *t* test was used for 2-group comparison; significant differences between multigroups were analyzed by 1-way ANOVA followed by the least significant difference

(LSD) test (equal variances assumed) or Welch ANOVA followed by Games–Howell multiple comparisons test (equal variances not assumed) at $P < 0.05$ level. PCA was used to distinguish the CHC profiles of AD1 and AD6 cockroaches.

## Supporting information

**S1 Fig. CHC profiles of 4-day-old sixth-instar nymphs of *Blattella germanica*.** Peak 24 represents the female-enriched 3,7-; 3,9-; and 3,11-DimeC29, and Peak 17 is the male-enriched 9-; 11-; 13-; and 15-MeC29. The data underlying this figure are included in S2 Data. CHC, cuticular hydrocarbon.
(TIF)

**S2 Fig. RNAi efficiency of different *BgElo* genes in *Blattella germanica*.** Data are shown as mean ± SEM, calculated from 3 to 4 replicates (2–3 cockroaches/replicate); $^{**}P < 0.01$, 2-tailed Student $t$ test. The data underlying this figure are included in S2 Data. RNAi, RNA interference.
(TIF)

**S3 Fig. Effects of *BgElo*-RNAi on CHCs of *Blattella germanica* with chain length longer than 30.** Different letters indicate significant differences between groups using Welch ANOVA (Games–Howell multiple comparisons test, $P < 0.05$). The data underlying this figure are included in S2 Data. CHC, cuticular hydrocarbon; RNAi, RNA interference.
(TIF)

**S4 Fig. Effects of *BgElo12*-RNAi and *BgElo24*-RNAi on CHC profiles of *B. germanica*. (A, B)** Relative amount of individual CHCs in males after knockdown of *BgElo12* and *BgElo24*. Data are shown as mean ± SEM; $^{*}P < 0.05$, $^{**}P < 0.01$; 2-tailed Student $t$ test, $n = 9$ or 10. **(C)** PCA of CHC profiles after repression of *BgElo12* and *BgElo24*; each dot represents a datum calculated from one cockroach. **(D)** The corresponding loading diagram; the numeric sequence labels correspond to numbers and CHC components in S2 Data. The data underlying this figure are included in S2 Data. CHC, cuticular hydrocarbon; PCA, principal component analysis; RNAi, RNA interference.
(TIF)

**S5 Fig. Verifying the roles of *BgElo12* and *BgElo24* in HC biosynthesis by the second RNAi targets. (A)** Analysis of CHCs after RNAi of *BgElo12* using the second target (ds*BgElo12B*) **(B)** or after RNAi of *BgElo24* using the second target (ds*BgElo24B*). **(C, D)** The effects of *BgElo12*-RNAi and *BgElo24*-RNAi on internal HCs. Data are shown as mean ± SEM; $^{*}P < 0.05$, $^{**}P < 0.01$; 2-tailed Student $t$ test, $n = 9$–12. The data underlying this figure are included in S2 Data. CHC, cuticular hydrocarbon; HC, hydrocarbon; RNAi, RNA interference.
(TIF)

**S6 Fig. Tissue-specific expression of *BgElo12* and *BgElo24* in *Blattella germanica*.** Data are shown as mean ± SEM; and each sample was collected from 4 (AC, Th, and Gu), 8 (FB, He, Ov, CG, and ED), and 12 (MT) cockroaches. Different letters indicate significant differences between groups using Welch ANOVA (Games–Howell multiple comparisons test, $P < 0.05$), $n = 4$. The data underlying this figure are included in S2 Data. AC, abdominal cuticle; Cg, colleterial gland; ED, ejaculatory duct; FB, fat body; Gu, gut; He, head; MT, Malpighian tubule; Ov, ovaries; Th, thorax.
(TIF)

**S7 Fig. Heterologous expression of *BgElo12* and *BgElo24* in yeast.** RNAi of *BgElo24* up-regulated the expression of *BgElo12* (**A**), while RNAi of *BgElo12* did not affect *BgElo24* transcript level (**B**). Data are shown as mean ± SEM; *P* values were calculated from 4 samples; each sample contained 2 cockroaches; 2-tailed Student *t* test. (**C**) Detection of GFP protein in the yeast with pYES2-GFP using a FV3000 confocal fluorescence microscope (Olympus). (**D, E**) RT-PCR analysis of the *BgElo12* and *BgElo24* mRNA after the induction with galactose. (**F, F', F"**) Gas chromatograms of FAMEs after adding 2-MeC16:0 into the medium. (**G, G', G"**) Gas chromatogram of FAMEs after adding 14-MeC16:0 into the medium. The compositions with retention times between 12 and 28 minutes were magnified about 50 times. (**H**, **I**) Mass spectra of methyl octacosanoate and methyl triacontanate, both of which showed a strong characteristic ion fragment (m/z = 74) and M peak. The data underlying S7A and S7B Fig are included in S2 Data. FAME, fatty acid methyl ester; RNAi, RNA interference; RT-PCR, real-time PCR.
(TIF)

**S8 Fig. The sex-specific developmental functions of *BgTra* and *BgDsx* in *Blattella germanica*.** RNAi of *BgTra* in females generated a male-like body size and cuticle color, male-like tergal gland structure, and a protruding tissue at the end of the abdomen (left center); RNAi of *BgDsx* in males generated a female-like body color and a protruding tissue at the end of the abdomen, and the tergal gland partly disappeared (right bottom). Other treatments did not generate obvious developmental effects. RNAi, RNA interference.
(TIF)

**S9 Fig. Transcriptional activity of *BgDsx^M* on the upstream regulatory sequence of *BgElo12*.** Data are shown as mean ± SEM; *P* values were calculated from 12 replicates; 2-tailed Student *t* test. The data underlying this figure are included in S2 Data.
(TIF)

**S10 Fig. Effects of *BgDsx*-RNAi in males and *BgTra*-RNAi in females on *BgElo24* expression.** Data are shown as mean ± SEM; *P* values were calculated from 4 replicates (2 cockroaches/replicate); 2-tailed Student *t* test. The data underlying this figure are included in S2 Data. RNAi, RNA interference.
(TIF)

**S11 Fig. Regulation of cuticular hrdrocarbon profiles by sex differentiation genes.** Data are shown as mean ± SEM, $^*P < 0.05$, $^{**}P < 0.01$, 2-tailed Student *t* test, $n = 14$ (ds*Muslta*-female), 12 (ds*BgTra*-female), 14 (ds*BgTra*+ds*BgDsx*-female), 12 (ds*Muslta*-male), and 16 (ds*BgDsx*-male). The data underlying this figure are included in S2 Data.
(TIF)

**S12 Fig. Regulation of internal hrdrocarbon profiles by sex differentiation genes. (A, B)** Effects of *BgTra*-RNAi in females and *BgDsx*-RNAi in males on sex-specific internal HC profiles. (**C, D**) Proportion changes of representative internal HCs after RNAi of *BgTra* in females and *BgDsx* in males. (**E, F**) Effects of *BgTra*-RNAi in females and *BgDsx*-RNAi in males on total amounts of internal HCs. Data are shown as mean ± SEM, $^*P < 0.05$, $^{**}P < 0.01$, 2-tailed Student *t* test, $n = 10$ or 11. The data underlying this figure are included in S2 Data. HC, hydrocarbon; RNAi, RNA interference.
(TIF)

**S1 Table. Quantification of difference CHCs during sexual dimorphic CHCs generation.** CHC, cuticular hydrocarbon.
(DOCX)

**S2 Table. Quantification of individual CHC after RNAi of other BgElo genes.** CHC, cuticular hydrocarbon; RNAi, RNA interference.
(DOCX)

**S3 Table. Calculation the proportions of different FAMEs in yeast expression.** FAME, fatty acid methyl ester.
(DOCX)

**S4 Table. Primer sequences used in this study.**
(DOCX)

**S1 Appendix. Sequence alignment and protein structure analysis of BgElos.**
(DOCX)

**S1 Data. The data underlying main figures (from Figs 1B–1E, 2A–2F, 3B–3H, 5A, 5C–5F', 6A–6D, and 6F–6G).**
(XLSX)

**S2 Data. The data underlying Supporting information figures (from S2, S3, S4A, S4B, S5A–S5D, S6A, S6B, S7A, S7B, S9, S10, S11A, S11B, S12A, S12B, S12C, S12D, S12E, and S12F Figs).**
(XLSX)

## Acknowledgments

We thank Shang-Wang Hong from Northwest A & F University for his help in sequence analysis and Hao-Su Cong and Professor Henry Chung from Michigan State University for their valuable suggestions.

## Author Contributions

**Conceptualization:** Xiao-Jin Pei, Yong-Liang Fan.

**Funding acquisition:** Yong-Liang Fan, Coby Schal, Tong-Xian Liu.

**Investigation:** Xiao-Jin Pei, Yu Bai, Tian-Tian Bai, Zhan-Feng Zhang, Nan Chen.

**Methodology:** Xiao-Jin Pei, Yu Bai, Zhan-Feng Zhang, Nan Chen.

**Project administration:** Yong-Liang Fan, Coby Schal, Tong-Xian Liu.

**Resources:** Yong-Liang Fan, Coby Schal, Sheng Li, Tong-Xian Liu.

**Supervision:** Yong-Liang Fan, Tong-Xian Liu.

**Writing – original draft:** Xiao-Jin Pei.

**Writing – review & editing:** Xiao-Jin Pei, Yong-Liang Fan, Yu Bai, Tian-Tian Bai, Coby Schal, Zhan-Feng Zhang, Nan Chen, Sheng Li, Tong-Xian Liu.

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
