## [Editor Report · Decision Letter 0]

1 Apr 2021

Dear Dr Liu, 

Thank you for submitting your manuscript entitled "Modulation of fatty acid elongation sustains sexually dimorphic hydrocarbons and female attractiveness in Blattella germanica (L.)" for consideration as a Research Article by PLOS Biology.

Your manuscript has now been evaluated by the PLOS Biology editorial staff, as well as by an academic editor with relevant expertise, and I'm writing to let you know that we would like to send your submission out for external peer review.

Please re-submit your manuscript within two working days, i.e. by Apr 05 2021 11:59PM.

Kind regards,

Roli Roberts

Senior Editor

PLOS Biology

---

## [Decision Letter · Decision Letter 1]

26 May 2021

Dear Dr Liu,

Thank you very much for submitting your manuscript "Modulation of fatty acid elongation sustains sexually dimorphic hydrocarbons and female attractiveness in Blattella germanica (L.)" for consideration as a Research Article at PLOS Biology. Your manuscript has been evaluated by the PLOS Biology editors, an Academic Editor with relevant expertise, and by four independent reviewers.

You'll see that all of the reviewers are positive about your study, and only have a few requests (mostly textual and presentational, but several new analyses are also required). In light of the reviews (below), we are pleased to offer you the opportunity to address the comments from the reviewers in a revised version that we anticipate should not take you very long. We will then assess your revised manuscript and your response to the reviewers' comments and we may consult the reviewers again.

IMPORTANT:

a) Please change your title to "Modulation of fatty acid elongation in cockroaches sustains sexually dimorphic hydrocarbons and female attractiveness"

b) Please provide a blurb, according to the instructions in the submission form.

c) When revising your manuscript, please bear in mind the need to make the paper accessible to our broad readership (for example, we like reviewer #2’s suggestion that you depict the biosynthetic pathway in one of the Figures).

d) Please attend to all of the requests from the reviewers.

e) Please attend to my Data Policy requests further down. Specifically, we note that you have provided some of the underlying data in the Supplementary Data files, but we think that data for some of the panels have not been presented separately. In addition, you should clearly cite the location of the data in each relevant Figure legend.

We expect to receive your revised manuscript within 1 month.

**IMPORTANT - SUBMITTING YOUR REVISION**

*Resubmission Checklist*

*Published Peer Review*

*PLOS Data Policy*

*Blot and Gel Data Policy*

Sincerely,

Roli Roberts

Roland Roberts

Senior Editor

PLOS Biology

rroberts@plos.org

DATA POLICY:

We note that you already present most of your underlying data in the supplementary data files S1 Data and S2 Data However, several panels are missing: Figs 1FG, 2CDEF, 3DE, 5CDEE’FF’, 6FG, S2, S3, S7AB, S10, S12CD. I suspect that these overlap with data presented for other panels, but please could you present it separately? NOTE: the numerical data provided should include all replicates AND the way in which the plotted mean and errors were derived (it should not present only the mean/average values).

We require the original, uncropped and minimally adjusted images supporting all blot and gel results reported in an article's figures or Supporting Information files. We will require these files before a manuscript can be accepted so please prepare and upload them now. Please carefully read our guidelines for how to prepare and upload this data: https://journals.plos.org/plosbiology/s/figures#loc-blot-and-gel-reporting-requirements 

DATA NOT SHOWN?

REVIEWERS' COMMENTS:

Reviewer #1:

This is a very well written manuscript that describes fatty acyl-CoA elongases and their regulation of elongation and pheromone production of hydrocarbon related pheromones in the German cockroach. Very little is known of the elongation process in producing hydrocarbons in insects, and this manuscript addresses this void. It is my opinion that it describes very careful and well done experiments.

Minor suggestions. 

Line 25. Correct the spelling of /underling'.

Line 70. I think that it would be useful to describe in a little more detail that the elongase they are looking at is the first step and rate limiting step in elongation, which is then followed by they three steps indicated, reduction of the ketone to an alcohol, dehydration of the alcohol to the alkene and then reduction of the alkene to a saturated fatty acyl-CoA.

Line 422. ....only n-alkanes and... with the n italicized.

Line 479. One of the best examples of hormonal regulation of chain length of hydrocarbons is the housefly, where ecdysone results in C27 alkenes being produced to where C23 alkenes become the dominant alkene. 

Reviewer #2:

This paper addresses the biosynthetic pathway and genetic mechanisms controlling sexual differentiation of cuticular hydrocarbon profiles in the German cockroach, Blatella germanica. The authors characterize two elongases involved in cuticular hydrocarbon (CHC) production and identify one that appears to underlie sexually dimorphic profiles and is under control of the tra/ dsx sex determination pathway. Manipulation of the elongase, dsx, and tra shift the CHC profile and results in masculinization of female CHCs. Overall, a tremendous amount of work went into this manuscript. The CHC measurements and in-vitro characterization of the elongase activities are convincing. The authors show interesting behavioral and physiological consequences resulting from knockdown of the elongase. The CHC and gene expression quantitation is carefully done and the proposed pathway is supported by the data. 

Despite the overall excellence of the technical work, some of the data need to be reanalyzed with proper statistical tests and more details are needed about the methods. The significance of the findings needs to be more explicitly stated for a general science audience. The role of the sex determination pathway in sexually dimorphic morphological features is well-documented. Aside from Drosophila, are there other examples of the SD cascade in controlling CHCs or pheromones? This manuscript would be greatly improved with the editing of a native English speaker. In addition, if this article is meant for a broad audience, in its current form the text is rather dense. There are also some experiments and observations included that are somewhat tangential and should be omitted (details below)

Minor corrections

61: Actually, the enzymatic pathways are well-characterized in multiple insect orders, as evidenced by the references provided in the rest of the paragraph and multiple reviews (Howard and Blomquist, 2005; Chung and Carroll, 2015; Blomquist and Bagneres, 2010; Martin and Drijfhout 2009; Yew and Chung 2015). Authors should consider including a pathway illustrating the biosynthesis of HCs to accompany this portion of the introduction.

74: replace "genes" with "enzymes". Again, I take issue with the statement that the genes/ enzymes are not well-studied. fact that the genes have been studied in few inspect species.

99-103: section on courtship can be shortened since this is discussed in detail in the results section.

172: "RNAi of the genes" is lab jargon. I suggest replacing "RNAi" with "knockdown"

220: N6D4: Please define in the text. It is only explained in Fig 6 legend.

228-229: awkwardly worded - "contributes to" in place of "support"? Also, the statement that "the role of Elo24 is desication tolerance" is overly general, based on a single assay. The enzyme may contribute to desiccation tolerance but it likely has other functions.

246: It looks like HCs in the C27 - C32 range were downregulated, not just C28-29.

255-256: Sentence could be reworded to make it more explicit what GFP expression indicates.

308 - 317: This section could be shortened and the explanation of age used for assays can go in methods

370-383: The observations about internal vs cuticular HC levels in this section are interesting but confusing in light of results showing that BgElo12 RNAi does not impact HC transport. The findings are also tangential to the main results and the underlying mechanisms can only be speculated upon at this point. I suggest drastically shortening this section, moving to the discussion section, or removing. 

453-455: Sentence construction is awkward.

478-479: The 20-HE steroid dehydrogenase was also identified as a regulator of HC profiles:

Chiang YN, Tan KJ, Chung H, Lavrynenko O, Shevchenko A, Yew JY. Steroid Hormone Signaling Is Essential for Pheromone Production and Oenocyte Survival. PLoS Genet. 2016 Jun 22;12(6):e1006126. doi: 10.1371/journal.pgen.1006126. PMID: 27333054; PMCID: PMC4917198.

488-489: Sentence construction is awkward.

494-497: I suggest removing text related to krh1. If authors think this is an important result, it should go into results section. However, the data provided are not sufficient to support the conclusion in 498-499.

Methods

More details are needed on the following methods:

1. How were the identifies of chromatogram peaks identified?

2. How was quantitation of GCMS peaks performed? The numbers provided in S1 and S2 tables are presented as absolute quantities. However, absolute quantification would require comparing the intensity of the peaks to a synthetic version of the compound. It does not appear that synthetic standards were used. The tables and figures should indicate that "relative quantities" are measured rather than absolute amounts. More details are also needed as to how percentage (e.g., 1B-E; 3D, E) was calculated. Is it percentage of all peaks detected or just percentage of HCs? 

3. Internal HC extractions: the fat body is diffuse throughout the body of the insect. How was it dissected in a standardized way from individual to individual? The reference (Fan et al.,JEB 2008) did not provide more details. 

Figures and Tables

S1, S2 table: "ug" in the footnote should be replaced with "μg". Please add sample size to footnote. If these numbers were calculated based on the relative intensity of the compound to the internal standard, they are relative quantities rather than absolute quantities. Table title should be "Relative quantification…"

S4, 5: See comment for Fig 3. 

S6 7A: What is the unit of y-axis? 

Fig 3B, C: An ANOVA is needed here to compare the relative amounts of individual CHCs rather than multiple t-tests. Before ANOVA can be performed, the CHCs should be collapsed into fewer categories so that the number of categories is less than the sample size. Y-axis should indicate that these are relative quantities.

Fig 3D: comparison of each CHC type between 3 different groups requires an ANOVA rather than multiple t-tests. 

Fig 5A: Was the C29 methyl ketone quantified using a synthetic standard or do numbers represent abundance relative to internal standard? 

Fig 6F: ANOVA should be used to compare between the 3 genotypes. 

Reviewer #3:

This study by Pei and co-workers addresses the interesting question of how sexually-dimorphic traits can emerge in animals, even though the genomes of males and females are essentially the same. In insects, cuticular hydrocarbons play a dual role. On the one hand, they protect from desiccation. On the other hand, they have in many cases been co-opted to play a role in chemical communication. Using the German cockroach, Blatella germanica, the authors unravel the molecular mechanism responsible for the formation of sexually-dimorphic cuticular hydrocarbons. Specifically, they identify elongase genes as major modulators of cuticular hydrocarbon composition. The study is thorough, from the characterization of sexually dimorphic cuticular hydrocarbons and their temporal development in both sexes to the identification of members of the multigene family encoding elongases (the enzymes catalyzing the extension of the fatty acid precursors). Furthermore, the authors elegantly explore the behavioral consequences of altering a single gene abundance via RNA interference. Finally, they built on previous work by Wexler et al (2019; Elife) and begin to explore how the sex-differentiation genes Transformer and Doublesex act as upstream regulators of BgElo12. 

All in all, this is a very interesting and complete study, with nice illustrations and clearly-presented results and conclusions. I have only a few small comments. 

In the RNAi efficiency experiments, it is not clear which tissue was used to performed these experiments. 

Lines 185-186: the authors wrote "we cannot rule out that other BgElo genes might have a less prominent role in HC biosynthesis". I think it is necessary to specify that this statement applies to the very long chain hydrocarbons (C27-C30), to reflect the data presented in the paper. 

Figure 2: while there is nothing wrong with using the Games-Howell multiple comparisons test, I think it would be more appropriate to use Dunnet's procedure and test dsBgEloX treatment vs dsMuslta control. The conclusions would most certainly not be affected. 

Figure 3: It looks like knocking down Elo24 is leading to less CHCs overall. It might be useful to visualize the data using percentages (absolute quantities). Also, I think it may be helpful to provide a PCA with the different treatments. Similarly to what authors do in Fig 1, a PCA plot would help the readers see the similarities and differences between female and male profiles and the impact of the treatments. 

Reviewer #4:

Pei et al. report the molecular mechanism underpinning the generation of sexually dimorphic hydrocarbons in the German cockroach. Expression analysis and RNAi screen against members of the fatty acid elongase gene family together with gene assays in a heterologous yeast system clearly demonstrated that both BgElo12 and BgElo24 are involved in the production of the hydrocarbons but only the female-enriched BgElo12 is responsible for sustaining the female-specific hydrocarbon profile. Furthermore, they reveal the involvement of two expression factors modulating the asymmetric expression of BgElo12 between the sexes.

The impressive amount of work results in an almost TOO massive manuscript, including 88 references, 12 supplementary figures, 4 supplementary table, one appendix and two data supplements. My focus has been on the main text and its 7 figures.

The study is very thorough, it addresses important scientific questions and the findings are of general interest. The manuscript is very well written and the conclusions supported by the results. 

The biosynthetic reaction under detailed study is the elongation of fatty acids, acids involved as precursors in the biosynthesis of the hydrocarbons produced from these fatty acids, hydrocarbons subsequently serving as precursors for the actual pheromone. 

- Can the authors explain and rectify why they did not analyze the effects of the elongases on the fatty acid profiles but only the resulting hydrocarbon profiles? 

- This appears especially important as they could not demonstrate the activity of BgElo12 and BgElo24 on the elongation of the methyl-branched fatty acids that have to be involved in the pheromone biosynthesis (cf. Lines 280-281). 

My additional few comments are all very minor:

Line 25: The sentence starting "Here, we report…." Does not read well to me. English is not my native language but I wonder if "underling" is really the right word? I would suggest "underpins".

Line 28: I would suggest that the sentence is slightly rephrased: "RNAi screen against the fatty acid elongase family members combined with heterologous expression of the genes in yeast revealed that both…."

Line 412: Delete "were"?

Line 505-506: While this is obviously true for moths (Lepidoptera) I wonder if it is generally true? Can the authors provide a reference?

Fig 1: Black=Male and Red=Female seems to apply to all panels but G. I would suggest the authors consider changing the colour code in G and possibly move the Black/Red -Male/Female explanation out of the "Frame A" as it applies to all of the other panels.

Fig 2: What is the unit on y-axis in A and B?

Fig 3: How do I find out what the different peak numbers stand for in A, B, C, D and E?

---

## [Editor Report · Decision Letter 2]

18 Jun 2021

Dear Dr Liu,

On behalf of my colleagues and the Academic Editor, Richard Benton, I'm pleased to say that we can in principle offer to publish your Research Article "Modulation of fatty acid elongation in cockroaches sustains sexually dimorphic hydrocarbons and female attractiveness" in PLOS Biology, provided you address any remaining formatting and reporting issues. These will be detailed in an email that will follow this letter and that you will usually receive within 2-3 business days, during which time no action is required from you. Please note that we will not be able to formally accept your manuscript and schedule it for publication until you have made the required changes.

IMPORTANT: The academic editor noticed that in Figure 3D, where you show "percentage changes", the y-axis runs from 0 to 1.0 (as if the value were fractions not percentages); if you really meant that the y-axis runs from 0 to 100%, then we recommend that you modify the labelling here (and possibly in other figure panels, if applicable).

PRESS: We frequently collaborate with press offices. If your institution or institutions have a press office, please notify them about your upcoming paper at this point, to enable them to help maximise its impact. If the press office is planning to promote your findings, we would be grateful if they could coordinate with biologypress@plos.org. If you have not yet opted out of the early version process, we ask that you notify us immediately of any press plans so that we may do so on your behalf.

Sincerely,

Roli Roberts

Roland G Roberts, PhD 

Senior Editor 

PLOS Biology

rroberts@plos.org